# GROUNDING CONTINUOUS REPRESENTATIONS IN GEOMETRY: EQUIVARIANT NEURAL FIELDS

**David R. Wessels**[*,1], **David M. Knigge**[*,2], **Riccardo Valperga**[2], **Samuele Papa**[2],
**Sharvaree Vadgama**[1], **Efstratios Gavves** [2,3], **Erik J. Bekkers** [1]
[1]AMLab, [2]VISLab, University of Amsterdam,
[3]Archimedes/Athena RC
`d.r.wessels@uva.nl, d.m.knigge@uva.nl`

## ABSTRACT

*Conditional Neural Fields* (CNFs) are increasingly being leveraged as continuous signal representations, by associating each data-sample with a latent variable that conditions a shared backbone Neural Field (NeF) to reconstruct the sample. However, existing CNF architectures face limitations when using this latent *downstream* in tasks requiring fine-grained geometric reasoning, such as classification and segmentation. We posit that this results from lack of explicit modelling of geometric information (e.g. locality in the signal or the orientation of a feature) in the latent space of CNFs. As such, we propose Equivariant Neural Fields (ENFs), a novel CNF architecture which uses a geometry-informed cross-attention to condition the NeF on a geometric variable—a latent point cloud of features—that enables an *equivariant* decoding from latent to field. We show that this approach induces a *steerability* property by which both field and latent are grounded in geometry and amenable to transformation laws: if the field transforms, the latent representation transforms accordingly—and vice versa. Crucially, this equivariance relation ensures that the latent is capable of (1) *representing geometric patterns faitfhully*, allowing for geometric reasoning in latent space, (2) *weight-sharing over similar local patterns*, allowing for efficient learning of datasets of fields. We validate these main properties in a range of tasks including classification, segmentation, forecasting, reconstruction and generative modelling, showing clear improvement over baselines with a geometry-free latent space. *Code attached to submission* here. *Code for a clean and minimal repo* here.

## 1 INTRODUCTION

Neural Fields (NeFs) (Xie et al., 2022) have recently gained prominence in the machine learning community as a novel representation method that models data as continuous functions. These fields, expressed as $f_\theta : \mathbb{R}^d \to \mathbb{R}^c$, map spatial coordinates—such as pixel locations $x \in \mathbb{R}^2$—to a corresponding signal, like RGB values $f_\theta(x) \in \mathbb{R}^3$, with $\theta$ representing the model's parameters. The parameters are optimized to approximate a target signal $f$, ensuring $\forall x : f(x) \approx f_\theta(x)$. This capability makes NeFs effective for representing continuous spatial, spatio-temporal, and geometric data, particularly in cases where grid-based methods fall short (Dupont et al., 2022). Their promise lies in serving as resolution-free representation that may be used irrespective of data resolution or discretization (Xie et al., 2022). Moreover, NeF-representations unify downstream models over different data modalities, allowing for transfer of modelling principles between data modalities that otherwise require data-specific engineering efforts (Dupont et al., 2022; Papa et al., 2023).

Building on this concept, *conditional neural fields* (CNFs) introduce a conditioning variable $z \in \mathcal{Z}$ to the model. Given a dataset of $N$ fields $\mathcal{D} = \{f_i : \mathbb{R}^d \to \mathbb{R}^c\}_{i=1}^N$, each specific field can now be represented by a specific conditioning variable $z_i$ via $\forall x : f_i(x) \approx f_\theta(x; z_i)$, while model weights $\theta$ are shared over the entire dataset. This approach enables CNFs to efficiently model datasets of fields using a set of latent variables that condition a *shared backbone* NeF. This allows for representing and analysing fields $f_i$ by means of their latent representation $z_i$, enabling novel approaches for

---

* Equal contribution.

Figure 1: Equivariant Neural Fields (ENFs) ground Neural Fields (NeFs) in geometry using a latent point cloud. A latent set $z$ consisting of tuples $(p_i, \mathbf{c}_i)$ of *pose* information $p_i$ and *context* $\mathbf{c}_i$ is optimized to reconstruct to the field $f(\cdot)$ as a function $f_\theta(\cdot; z)$ using gradient-descent. Due to their explicit positional grounding and locality, the latent retains important geometric features in the input field. The latent $z$ can then be used in downstream tasks, e.g. classification, segmentation, and geometric reasoning, where transformations in the field are mirrored in the latent representation through group actions $L_g[f] \sim g \cdot z$.

solving tasks involving fields through a framework known as *learning with functa* (Dupont et al., 2022). Applications of this include tasks such as classification, segmentation, and the generation of continuous fields (Dupont et al., 2022; Papa et al., 2023), as well as continuous PDE forecasting by solving dynamics in the latent space (Yin et al., 2022; Knigge et al., 2024).

**Geometry in CNFs** A notable limitation of conventional CNFs, however, is a lack of explicit geometric interpretability; each field $f_i$ is encoded by a "global" variable $z_i$, meaning for instance that any notion of locality or other explicit spatial relationships—which have proven a strong inductive bias in computer vision—is lost. Although this global representation has inherent benefits, e.g. enabling the use of simple MLPs as downstream models and allowing for intuitive interpolation between latent signal representations, empirically it has shown limited performance in settings where samples of the dataset are not consistently globally aligned (Bauer et al., 2023). For instance, in classification tasks, spatial organisation of an image's content is crucial for understanding shape and enabling geometric reasoning (Van Quang et al., 2019); current neural fields lack such geometric inductive biases, limiting their performance in e.g. classification and generative modelling (Bauer et al., 2023; Papa et al., 2023). To this end, we propose *equivariant neural fields* (ENFs), a new class of NeFs that allows for the identification of continuous fields with concrete geometric representations (Fig. 1).

**Geometry-grounded neural fields** When the goal is to utilise field representations $z_i$ in downstream tasks, it is crucial that these representations capture both textural/appearance information and explicit geometric information. Our approach is inspired by the idea of *neural ideograms*—learnable geometric representations (Vadgama et al., 2022; 2023), and addresses the pervasive issue of *texture bias*, which causes typical deep learning systems to overfit to textural patterns and ignore important geometric cues (Geirhos et al., 2018; Hermann et al., 2020). To address this challenge, we propose defining representations that (1) explicitly separate aspects of geometry–specifically the *pose* of features–from appearance and (2) are localized in the input signal such that geometric concepts like orientation and distance are maintained from input to latent space. This necessitates that the geometric components of the representations have a meaningful structure, adhering to the same group transformation laws applicable to the fields. Geometrically, this means that distortions in the field translate to corresponding distortions in the latent space, ensuring that geometric (shape) variations are preserved and representable in latent space.

To establish an explicit grounding in geometry, we propose modelling the conditioning variables as geometric point clouds $z = \{(p_i, \mathbf{c}_i)\}_{i=1}^N$, comprising $N$ pose-appearance tuples, with $p_i \in G$ being a pose (element) in a group $G$, and $\mathbf{c}_i \in \mathbb{R}^c$ an appearance vector. This representation space has a well-defined group action, namely $gz = \{(gp_i, \mathbf{c}_i)\}_{i=1}^N$, allowing us to formalise the notion of

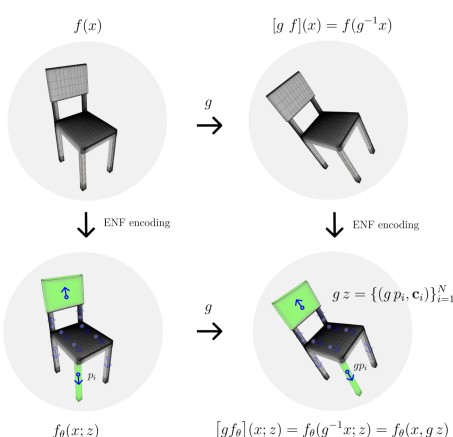

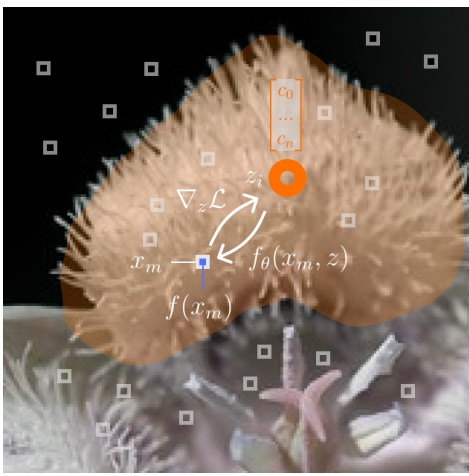

Figure 2: ENFs preserve transformations through their steerability property; if the field transforms with a group action $g$, the latents transform accordingly via the following group action on the pointcloud; $gz = \{gp_i, \mathbf{c}_i\}_{i=1}^N$.

Figure 3: ENFs are a local signal encoding; a latent $z_i$ is optimized to represent a local signal patch. We show that this inductive bias allows for leveraging weight-sharing, and improves downstream performance by retaining important geometric features.

grounding a neural field through the following

$$\text{Steerability property:} \qquad \forall g \in G : f_\theta(g^{-1}x; z) = f_\theta(x; gz). \tag{1}$$

This property ensures that if the field transforms, the latent transforms accordingly (Fig. 2). This property has also been shown in (Atzmon et al., 2022; Chatzipantazis et al., 2022) for learning implicit shape representations and is well known in equivariance literature more broadly (Bekkers, 2019; Cohen et al., 2019; Weiler & Cesa, 2019; Deng et al., 2021).

**Contributions** With this work we present the following contributions: A new class of geometry-grounded equivariant neural fields that posses

- the *steerability* property and thus proveable generalization over group actions
- *weight sharing* which enables more efficient learning
- *localized representations* in a latent point set which enables unique editing properties

We verify these properties through a range of experiments that (1) support the claim that latents are *geometrically meaningful*, (2) show competitive reconstruction and representation capacity on segmentation, classification, forecasting and super-resolution tasks on image and shape data.

## 2 BACKGROUND

**Neural Fields and conditioning variables** Neural Fields (NeFs) are learned functions $f_\theta$ mapping signal coordinates $x$ to signal values $f(x)$, parameterized by a neural network with parameters $\theta$. Due to their flexibility they have emerged as prominent continuous data representation, applied on datatypes varying from object or scene data (Park et al., 2019; Mescheder et al., 2019; Sitzmann et al., 2020a; Mildenhall et al., 2021) to audio and images Tancik et al. (2020); Sitzmann et al. (2020b). In order to more efficiently represent whole datasets of signals, Conditional Neural Fields only optimise a latent conditioning variable $z^f$ per signal-instance $f \in D$ instead of optimising a separate set of neural network parameters $\theta_i$. The two most common approaches for obtaining latents $z^f$ are autodecoding (Park et al., 2019) - where latents $z^f$ are initialized and optimized alongside NeF parameters $\theta$ - and Meta-Learning based encoding - where optimization is split into an outer loop that optimizes the backbone $\theta$ and an inner loop that optimizes latents $z^f$. We explain both in detail in Appx. A.1.1, and use both in experiments described in Sec. 4.

In seminal work by Dupont et al. (2022), a datatype agnostic approach for *learning* on these continuous signal representations was proposed - involving first the optimization of a set of conditioning variables $z$ to reconstruct a dataset of signals $\mathcal{D} := \{f_j\}_{j=1}^n \sim \{z^{f_j}\}_{j=1}^n$, and afterward using these variables as a surrogate for the data in downstream tasks such as classification, generation and completion. Although this work highlighted the flexible data-agnostic nature of Conditional Neural Fields (CNFs) by representing signals through a single "global" condition variable $z^{f_j}$, later work by Bauer et al. (2023) showed their limitations in performance for more complex tasks (i.e. involving higher-resolution more varied natural data).

**Group theoretical preliminaries**   The notion of transformation-preservation of an operator - such as the relation between field $f$ and latent $z$ - is best expressed through group theory. A group is an algebraic construction $(G, \cdot)$ defined by a set $G$ and a binary operator $\cdot : G \times G \to G$ called the *group product*, satisfying: *closure*: $\forall h, g \in G : h \cdot g \in G$, *identity*: $\exists e \in G : \forall g \in G, g \cdot e = g$, *inverse*: $\forall g \exists g^{-1} \in G : g \cdot g^{-1} = e$ and *associativity*: $\forall g, h, i \in G : (g \cdot h) \cdot i = g \cdot (h \cdot i)$

Given such a group $G$ with identity element $e \in G$, and a set $X$, we can define the *group action* as a map $G \times X \to X$, which we will denote with direct multiplication i.e. a group element $g \in G$ action on a coordinate $x \in X$ is denoted $gx$. Note that when $X = G$ the group action equals the group product. For the group action on fields $f : X \to \mathbb{R}$ we use a separate symbol, namely $[\mathcal{L}_g f](x) := f(g^{-1}x)$. In this work we are interested in the Special Euclidean group $SE(n) = T_n \rtimes SO(n)$. $SE(n)$ is the roto-translation group consisting of elements $g = (\mathbf{t}, \mathbf{R})$ with group operation $g\, g' = (\mathbf{t}, \mathbf{R})\,(\mathbf{t}', \mathbf{R}') = (\mathbf{t} + \mathbf{R}\mathbf{t}', \mathbf{R}\mathbf{R}')$; the left-regular action on function spaces is defined by $\mathcal{L}_g f(x) = f(g^{-1}x) = f(\mathbf{R}^{-1}(x - \mathbf{t}))$.

**Equivariant graph neural networks for downstream tasks.**   A key property of our framework is its ability to *associate geometric representations with fields*. This capability unlocks a rich toolset for field analysis through the lens of geometric deep learning (GDL) (Bronstein et al., 2021). The GDL field has made significant advancements in the analysis and processing of geometric data, including the study of molecular properties (Batzner et al., 2022; Batatia et al., 2022; Gasteiger et al., 2021; Brandstetter et al., 2021) and the generation of molecules (Hoogeboom et al., 2022; Bekkers et al., 2023) and protein backbones (Corso et al., 2022; Yim et al., 2023). In essence, these approaches *characterise shape*. Our encoding scheme makes these tools now applicable to analysing the geometric components of neural fields, shown in Sec. 4.

Equivariant Graph Neural Networks (EGNNs) are a class of Graph Neural Networks (GNNs) that imposes roto-translational equivariance constraints on their message passing operators to ensure that the learned representations adhere to specific transformation symmetries of the data. Among the various forms of equivariant graph NNs (Thomas et al., 2018; Brandstetter et al., 2021; Satorras et al., 2021; Gasteiger et al., 2021; Bekkers et al., 2023; Eijkelboom et al., 2023; Kofinas et al., 2024) we will utilise PθNITA (Bekkers et al., 2023) as an equivariant operator to analyse and process our latent point-set representations of fields. For the neural field, we leverage the same optimal bi-invariant attributes as introduced in Bekkers et al. (2023)—which are based on the theory of homogeneous spaces—to parameterise our neural fields, allowing for seamless integration. They formalise the notion of *weight sharing* in convolutional networks as the sharing of message functions (kernels) over point-pairs - e.g. relative pixel positions - that should be treated equally. By defining equivalence classes of point-pairs that are identical up to a transformation in the group, we too derive attributes that uniquely identify these classes and enable weight sharing in our proposed ENFs.

## 3   METHOD

In this section we introduce the Equivariant Neural Field (ENF) architecture. We start Sec. 3.1 by showing how we impose the proposed steerability property (Eq. 1), through the definition of *bi-invariant attributes*. We construct a cross-attention operator conditioned by these attributes. We subsequently constrain the operator to represent local sub-regions of the input domain by applying a Gaussian window. Finally, we propose a k-nearest neighbouring approach to cope with the computational complexity of the cross-attention operation between pixels and latents. In Sec. 3.2 we finally discuss how we obtain a latent point cloud $z^f$ for a signal $f$.

**Bi-invariance constraint**   Before presenting our equivariant neural field design, we need to understand the constraints imposed by the steerability property (Eq. 1). The key result–shown also by

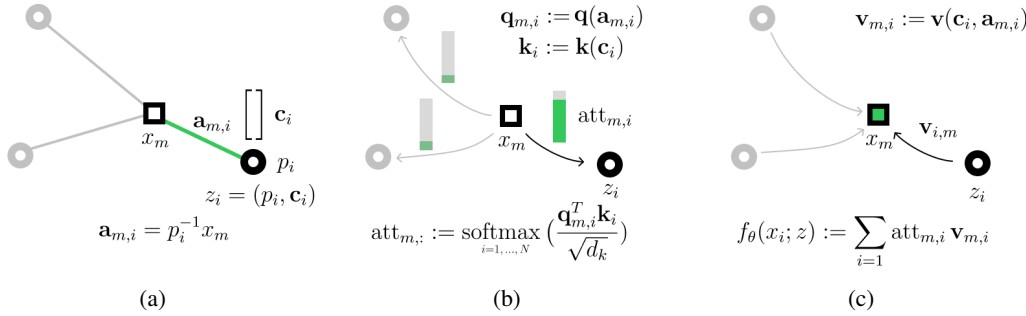

Figure 4: A visual intuition for the proposed cross-attention between coordinate $x_m$ and latent $z = \{(p_i, \mathbf{c}_i)\}_{i=1}^N$. (a) Bi-invariant $\mathbf{a}_{m,i}$ is calculated between coordinate $x_m$ and pose $p_i$ as $p_i^{-1}x_m$. (b) The query and key functions $\mathbf{q}$ transforms $\mathbf{a}_{m,i}$ into a query $\mathbf{q}_{m,i}$, and key function $\mathbf{k}$ maps context vector $\mathbf{c}_i$ to key $\mathbf{k}_i$. Attention coefficients are calculated through a softmax over $\mathbf{q}_{m,i}\mathbf{k}_i$. The softmax is taken over the $N$ latents, yielding $N$ attention coefficients $\text{att}_{m,i}$, one for each latent $z_i$. (c) A value $\mathbf{v}_{m,i}$ for each latent-coordinate pair is calculated as a function $v$ of $\mathbf{c}_i$ and $\mathbf{a}_i$ - and the resulting values are aggregated, weighted by their corresponding attention coefficients $\text{att}_{m,i}$.

Cohen et al. (2019); Bekkers (2019) in the context of equivariant CNNs–is that for steerability, the field $f_\theta$ must be bi-invariant with respect to both coordinates and latents.

**Lemma 1.** *A conditional neural field satisfies the steerability property iff it is bi-invariant, i.e.,* $\forall g \in G: \; f_\theta(gx; gz) = f_\theta(x; z)$.

*Proof.* If $f_\theta$ satisfies the steerability property, then $f_\theta(gx; gz) = f_\theta(g^{-1}gx; z) = f_\theta(x; z)$, so it is bi-invariant. Conversely, if $f_\theta$ is bi-invariant, then $f_\theta(g^{-1}x; z) = f_\theta(gg^{-1}x; gz) = f_\theta(x; gz)$, satisfying the steerability property equation 1. $\qquad\square$

### 3.1 EQUIVARIANT NEURAL FIELDS CONDITIONED ON GEOMETRIC ATTRIBUTES

The cross-attention operation enables the use of latent-sets as conditioning variables for CNFs by applying cross-attention between embedded coordinates $x_m$ and a latent set of context vectors $z = \{\mathbf{c}_i\}_{i=1}^N$ (Zhang et al., 2023). Such cross-attention fields assign to each $x_m$ a corresponding value $f_\theta(x_m; z)$, by matching a coordinate (query) embedding $\mathbf{q}(x_m)$ against latent (key) vectors $\mathbf{k}(\mathbf{c}_i)$ to obtain attention coefficients $\text{att}_{m,i}$, and aggregating associated values $\mathbf{v}(\mathbf{c}_i)$ via

$$f_\theta(x_m; z) = \mathbf{W}_o \sum_{i=1}^N \text{att}_{m,i}\, \mathbf{v}(\mathbf{c}_i) \qquad \text{with} \qquad \text{att}_{m,:} = \underset{i=1,\dots,N}{\text{softmax}}\left( \frac{\mathbf{q}(x_m)^T \mathbf{k}(\mathbf{c}_i)}{\sqrt{d_k}} \right),$$

where $\mathbf{W}_o$ maps cross-attention outputs to NeF output/signal dimension $\mathbb{R}^c$. Our desired latent representation contains a geometric component, namely the poses $p_i$ associated with the context vectors $\mathbf{c}_i$. In order to see how this geometric information could be leveraged, we highlight how each of the three components $(\mathbf{q}, \mathbf{k}, \mathbf{v})$ could depend on the geometric attributes:

$$f_\theta(x_m; z) = \mathbf{W}_o \sum_{i=1}^N \text{att}_{m,i}\, \mathbf{v}(x_m, p_i, \mathbf{c}_i) \qquad \text{with} \qquad \text{att}_{m,:} = \underset{i=1,\dots,N}{\text{softmax}}\left( \frac{\mathbf{q}(x_m, p_i)^T \mathbf{k}(x_m, p_i, \mathbf{c}_i)}{\sqrt{d_k}} \right).$$

The steerability condition demands that the field has to be bi-invariant with respect to transformations on both $x_m$ and $p_i$, and the easiest way to achieve this is to replace any instance of $x_m, p_i$ by an invariant pair-wise attribute $\mathbf{a}(x_m, p_i)$ that is both invariant and maximally informative. With maximally informative we mean that coordinate-pose pairs that are not the same up to a group action receive a different vector descriptor, i.e., $\mathbf{a}(x_m, p_i) = \mathbf{a}(x'_m, p'_i)$ *if and only if* there exists a $g \in G$ such that $x'_m = gx_m$ and $p'_i = gp_i$.

**Equivariant Neural Fields** Based on recent results in the context of equivariant graph neural networks Bekkers et al. (2023), we let $\mathbf{a}(x_m, p_i) := p_i^{-1}x$ be the the unique and complete *bijective* identifier for the equivalence class of all coordinate-pose pairs that are the same up to a group

action. Bijectivity here implies that the descriptor $p_i^{-1}x_m$ contains all information possible to identify the equivalence classes, and thus the use of those attributes leads to maximal expressivity. In this work we use bi-invariants for translation ($\mathbf{a}^{\mathbb{R}^n}$) roto-translation ($\mathbf{a}^{\mathrm{SE}(2)}$) - as well as a "bi-invariant" that breaks any equivariance $\mathbf{a}^{\emptyset}$ (see Appx. B for details). We define the ENF as follows:

$$f_\theta(x; z) = \mathbf{W}_o \sum_{i=1}^N \mathrm{att}_{m,i}\, \mathbf{v}(\mathbf{a}(x, p_i), \mathbf{c}_i) \qquad \text{with} \qquad \mathrm{att}_{m,:} = \underset{i=1,\dots,N}{\mathrm{softmax}} \left( \frac{\mathbf{q}(\mathbf{a}(x, p_i))^T \mathbf{k}(\mathbf{c}_i)}{\sqrt{d_k}} \right).$$

As specific parameterizations for $\mathbf{a}, \mathbf{q}, \mathbf{k}, \mathbf{v}$, we choose:

$$\mathbf{a}(x, p_i) := \phi(p_i^{-1}x) \tag{2}$$

$$\mathbf{q}(\mathbf{a}(p_i^{-1}x)) := \mathbf{W}_q \mathbf{a}(x, p_i) \qquad \mathbf{k}(\mathbf{c}_i) := \mathbf{W}_k \mathbf{c}_i, \tag{3}$$

$$\mathbf{v}(\mathbf{a}(x, p_i), \mathbf{c}_i) := (\mathbf{W}_v \mathbf{c}_i) \odot (\mathbf{W}_{a_\gamma} \mathbf{a}(x, p_i)) + (\mathbf{W}_{a_\beta} \mathbf{a}(x, p_i)) \tag{4}$$

with $\odot$ denoting element-wise multiplication and $\phi$ a relative coordinate embedding function which we set to be a Gaussian RFF embedding (Tancik et al., 2020). In neural field literature it is known that neural networks suffer from high spectral biases (Rahaman et al., 2019). Due to smooth input-output mappings it becomes difficult to learn high-frequency information in low-dimensions such as the coordinate inputs for a NeF. Gaussian RFF embeddings introduce high-frequency signals in the embedding to alleviate the spectral bias.

Since the value transform has as goal to fill in spatially varying signal patches during reconstruction, the value-function is also conditioned on the geometric attributes $p^{-1}x$. To add extra expressivity we chose to apply the conditioning via FiLM modulation (Perez et al., 2018) which applies a feature-wise linear modulation with a learnable shift $\beta$ and scale $\gamma$ modulation.

A crucial difference with standard transformer-type methods on point clouds is that cross-attention is between *relative position embeddings*—relative to the latent pose $p_i$—and that the value transform is of depth-wise separable convolutional form (Chollet, 2017; Bekkers et al., 2023), which is a stronger form of conditioning (Koishekenov & Bekkers, 2023) than additive modulation as is typically done in biased self-attention networks such as Point Transformer (Zhao et al., 2021).

**Enforcing and learning locality in the latent point cloud** With the current proposed setup, cross-attention is universally applied across the entire set of latents and coordinates. Given the Softmax distribution, each coordinate indiscriminately receives a nonzero attention value for every latent $(p_i, \mathbf{c}_i) \in z$. Consequently, although latents possess inherent latent space positional attributes, they do not strictly represent localised regions of the signal, breaking the locality we require.

To address this issue, we modify the attention mechanism by incorporating a Gaussian spatial windowing function with parameter $\sigma_{\mathrm{att}}$ into the computation of attention coefficients. This approach follows the strategy proposed by Cordonnier et al. (2019). Specifically, the attention scores derived from the dot product between the query and key values are modulated by a Gaussian window $\mu$, which is dependent on the Euclidean distance between latent positions $p^{pos}$ and the input coordinates, expressed as $\mu_\sigma(x_m, p_i) = -\sigma_{\mathrm{att}}||p_i^{pos} - x_m||^2$. Here, $\sigma_{\mathrm{att}}$ is a hyperparameter that regulates the size for each latent (Fig. 5). We have:

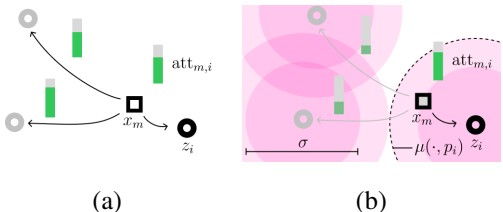

(a)      (b)

Figure 5: (a) Global attention between coordinates $x_m$ and latents $z_i$ can result in high attention values for non-local latents. (b) Locality is enforced through a Gaussian window $\mu_\sigma(x_m, p_i)$, attenuating the dot-product $\mathbf{q}_{m,i}\mathbf{k}_i$ as a function of the distance between $x_m$ and $p_i$.

$$att(x, z) = \underset{i=1,\dots,N}{\mathrm{softmax}} \left( \frac{\mathbf{q}(\mathbf{a}(x, p_i))^T \mathbf{k}(\mathbf{c}_i)}{\sqrt{d_k}} + \mu_\sigma(x, p_i) \right), \tag{5}$$

where $\mu(x_m, p_i)$ represents the Gaussian window computed for each latent position. To enhance the expressiveness of the model even further, $\sigma_{att}$ can be made latent-specific, encoding for the spatial extent of a latent $z_i$. This extension allows the latents to be expressed as point clouds: $z^f = \{(p_i, \mathbf{c_i}, \sigma_i)\}_{i=1}^N$, effectively coupling position, appearance, and locality attributes within the latent space. However, we keep this for further research - fixing $\sigma_{att}$ in our experiments.

**A note on efficiency**   A limitation of the proposed method is the considerable computational complexity required to calculate output values for large numbers of input coordinates; larger more complex signals require a larger number of latents to be represented accurately leading to an exponential number of attention coefficients needing to be calculated each forward pass (complexity scales as $O(N_{\text{latents}} \times N_{\text{coordinates}})$). Since we localize latents, larger input domains present a trade-off. Either we maintain a small number of latent points—requiring a larger $\sigma_{\text{att}}$ to cover the entire domain, diminishing locality—or we increase the number of latent points, aggravating computational cost.

To mitigate the computational overhead associated with the cross-attention between the latent points and the sampled coordinates, we propose employing a k-nearest neighbours (k-NN) approach for the cross-attention operation. Specifically, for each pixel, we first identify its k-nearest latent points and then compute cross-attention for this coordinate only over these $k$ nearest latents. This approach reduces computational cost while preserving the advantages of local representations (Appx. D.5).

**Properties of Equivariant Neural Fields**
*Weight sharing* - Conditioning ENF's attention operator on attributes **a** which uniquely identify equivalence classes of (latent-coordinate)-pairs, ensures that the cross-attention operators—be it the attention logits or the value transform—respond similarly regardless of the pose under which a signal pattern presents itself. This form of *weight-sharing* has shown improved data and representation efficiency in GNN-based architectures (Bekkers et al., 2023) and—as such—we hypothesize that our proposed ENF architecture similarly benefits from these properties compared to other types of NeFs. We confirm this property in figure 10, which shows that ENFs share weights over group actions $g \in G$ for the geometry in which

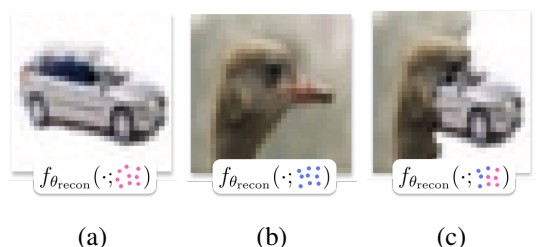

(a)          (b)          (c)

Figure 6: Latent point cloud editing. Subfigures (a) and (b) show two reconstructed CIFAR-10 images with corresponding latents $z^{\text{car}}, z^{\text{duck}}$. Subfigure (c) shows a reconstruction of the latent set $z^{\text{car-duck}}$ when selecting latents from either $z^{\text{car}}$ or $z^{\text{duck}}$ based on their position.

the point clouds are grounded. In Sec. 4 we verify these benefits on downstream tasks.

*Locality and (geometric) interpretability-* The use of latent point clouds allows for localization of the cross-attention mechanisms *around* the latent pose, akin to how a convolution operator works with localised kernels. Not only does this improve interpretability and downstream performance (field patterns can be attributed to specific latent points), it enables unique field editing possibilities. Since our method is based on sets of latents and each element is responsible for a local neighbourhood of the input domain, we can take arbitrary unions(e.g. stitching) or intersections(e.g. latent-merging) of point-sets of different samples (Fig. 6).

## 3.2   OBTAINING LATENT POINT CLOUDS $z$

Following Park et al. (2019); Dupont et al. (2022) we obtain a latent point cloud $z^f$ for a specific sample $f$ using *gradient descent*, optimizing $z^f$ for reconstruction of the original signal $f$. For instance, in images a latent point cloud $z$ may be optimized for an $L_2$-loss between $f_\theta(\cdot, z)$ and $f(\cdot)$. Of course, this also requires optimizing $\theta$ to meaningfully map latents to fields. The two most common approaches to this end are Autodecoding (Park et al., 2019) - where $z^f$ and $\theta$ are optimised simultaneously over a dataset, or MAML (Finn et al., 2017; Tancik et al., 2021) - where optimization is split into an outer and inner loop, with $\theta$ being optimized in the outer loop and $z$ being re-initialized every outer step to reconstruct the current signal batch in a limited number of SGD steps in the inner loop. We detail these approaches in Appx. A.1.1, using both in the experiments.

## 4   EXPERIMENTS

First, we show the ability of Equivariant Neural Fields (ENFs) to reconstruct datasets of input fields $f$ - i.e. to associate a latent point cloud $z^{f_j}$ to a given dataset of samples $f_j \in D$ that accurately reconstructs them. Then, we validate ENFs as an improved NeF-based *downstream representation*

Table 1: Test-set reconstruction PSNR (db↑) on CIFAR10, CelebA64x64, ImageNet1k, test accuracy (%↑) on CIFAR10.

|  | CIFAR10 | | CELEBA | IMAGENET |
|---|---|---|---|---|
| TASK | RECON. | CLASS. | RECON. | RECON. |
| Functa | 38.1 | 68.3 | 28.0 | 7.2 |
| ENF $\mathbf{a}^{\emptyset}$ | 36.5 | 68.7 | 30.6 | 24.7 |
| ENF $\mathbf{a}^{\mathbb{R}^2}$ | **42.2** | **82.1** | **34.6** | **27.5** |
| ENF $\mathbf{a}^{\mathrm{SE}(2)}$ | 41.6 | 81.5 | 32.9 | 26.8 |

Table 2: Test reconstruction (IoU↑) on ShapeNet16 and ShapeNet55 and test classification accuracy (%↑) on ShapeNet16.

| MODALITY | SHAPENET16 VOXEL (OCC) | | SHAPENET55 P. CLOUD (SDF) |
|---|---|---|---|
| TASK | RECON. | CLASS. | RECON. |
| NF2vec | - | 93.3 | - |
| Functa | 92.1 | 90.3 | 25.7 |
| ENF $\mathbf{a}^{\emptyset}$ | 90.7 | 96.4 | 72.3 |
| ENF $\mathbf{a}^{\mathbb{R}^3}$ | **92.9** | **96.6** | **73.2** |

for various tasks requiring geometric reasoning; classification, segmentation and forecasting. To show the flexibility of NeF-based representations, we perform these tasks on a range of modalities.

Each downstream experiment consists of two stages: (1) fitting a ENF backbone $f_\theta$ for *reconstruction* of the input fields $f_j \in D$ to obtain latents pointclouds $z^{f_j}$, and (2) training a downstream model—that takes $z^{f_j}$ as input—for each specific task. The bi-invariant $\mathbf{a}_{m,i}$ we choose to condition our ENF, as well as the downstream model, varies depending one the type of task we're performing—described per experiment below. Since our goal is to assess the impact of grounding continuous representations in geometry, besides dataset-specific baselines, we also compare against Functa (Dupont et al., 2022), the framework that originally proposed functional representations $z^{f_j}$ as data surrogates.

For experimental details and hyperparameters we refer to Appx. C.

## 4.1 RECONSTRUCTION CAPACITY

First we evaluate our proposed ENFs on their reconstruction capabilities. • **Image data** We show results for reconstruction trained with Meta-Learning on CIFAR10 Krizhevsky et al. (2009), CelebA64×64 (Liu et al., 2015) and ImageNet1K (Deng et al., 2009) using different bi-invariant attributes $\mathbf{a}_{m,i}$—resulting in equivariance to different corresponding transformation groups—in Tab. 1. We provide results for a Functa baseline, following the setup described in (Dupont et al., 2022). Notably, translational weight sharing ($\mathbf{a}^{\mathbb{R}^2}$) outperforms settings with no-transformation ($\mathbf{a}^{\emptyset}$) and roto-translational weight sharing ($\mathbf{a}^{\mathrm{SE}(2)}$). Moreover, it seems that locality alone is itself a useful inductive bias when moving to higher resolution, more varied images; Functa outperforms ENF $\mathbf{a}^{\emptyset}$ on CIFAR10 reconstruction, but on CelebA and ImageNet all ENF parameterizations outperform Functa. These results reinforce locality and equivariance as inductive biases in image-based continuous reconstruction tasks. • **Shape data** We show shape reconstruction results (Tab. 2) on two common shape representations; voxels (3D occupancy grids) and meshes. For voxel data we take train and test splits from the 16-class ShapeNet-Part segmentation dataset (Yi et al., 2016) (which we denote ShapeNet16) and fit their corresponding voxel-based representation as occupancy function $\mathbb{R}^3 \rightarrow \{0, 1\}$ - also using the obtained representations in the ShapeNet-Part segmentation experiment detailed below. For mesh data we opt instead to fit the full 55-class ShapeNetCore (v2) object dataset (Chang et al., 2015), fitting these with 150k points sampled from the signed distance function of the mesh (more details in Appx. C.3). Unlike Dupont et al. (2022), we were unable to get sufficient quality reconstructions with meta-learning and instead obtain latents $z$ using autodecoding on shape data (finding discussed in Appx. A.1.1). Results show that ours as well as the baseline models struggle with accurately reconstructing the underlying shape from the SDF point clouds, we think due to the more complex optimization objective.

## 4.2 DOWNSTREAM TASKS

**Image classification** One major limitation of Functa—noted by Bauer et al. (2023)—was lacking performance on complex image tasks such as classification. To show performance of our model in this setting, we reproduce the CIFAR10 classification experiment listed in Bauer et al. (2023)—augmenting CIFAR10 with 50 random crops and flips per image, and training an ENF to reconstruct these using meta-learning, obtaining latent point clouds $z$. We do this for different bi-invariants $\mathbf{a}$ corresponding to no equivariance ($\mathbf{a}^{\emptyset}$), translational ($\mathbf{a}^{\mathbb{R}^2}$) and roto-translational ($\mathbf{a}^{\mathrm{SE}(2)}$) equivari-

ance. We then train a PΘNITA classifier (Bekkers et al., 2023) to classify these latent point clouds - conditioning the message passing function on the same bi-invariants, now calculated between poses $p_i$. Results (Tab. 1) show a test-accuracy improvement of 13.8 percentage points (68.7%→82.1%) over Functa (Dupont et al., 2022), and also indicate that in this setting (roto-)translational equivariance is a strong inductive bias—with $\mathbf{a}^{\mathrm{SE}(2)}$, $\mathbf{a}^{\mathbb{R}^2}$-ENFs outperforming $\mathbf{a}^{\emptyset}$-ENFs.

**Shape classification** Highlighting the flexibility of NeF-representations we apply the same setup to shape classification, training PΘNITA classifiers on the aforementioned ShapeNet16 dataset. Results in Tab. 2 show that relevant geometric features are better preserved in a localized latent space. Here, the performance difference between equivariant ($\mathbf{a}^{\mathbb{R}^3}$) and non-equivariant ($\mathbf{a}^{\emptyset}$) ENFs are negligible. This is to be expected due to the global alignment of the ShapeNet dataset, and shows ENF is able to perform under equivariance constraints even in non-equivariant tasks.

**ShapeNet-Part segmentation** Where classification primarily evaluates how well the latent point clouds captures global information, we also want to evaluate ENFs performance on fine-grained tasks. As such, we evaluate on the ShapeNet part segmentation task (Yi et al., 2016). The ShapeNet-Part dataset consists of point clouds for 16 ShapeNet object-classes, each with a varying number of annotated parts for a total of 50 segmentation classes. We use the ENF $\mathbf{a}^{\mathbb{R}^3}$ backbone defined in the voxel reconstruction task above ($f_{\theta_{\mathrm{recon}}}$) to obtain a latent $z^{\mathrm{recon}}$ for a shape. Then, a second ENF ($f_{\theta_{\mathrm{seg}}}$) is trained to map points on this shape to a one-hot encoding of their corresponding segmentation classes, i.e. $f_{\theta_{\mathrm{seg}}}(x_m; z^{\mathrm{recon}})$ maps $x_m$ to its class label $y_m$. Results (Tab. 3) somewhat surprisingly show ENF and Functa perform

Table 3: Mean class and instance IoU (↑) on ShapeNet.

| MODEL | INST. MIOU | CLS. MIOU |
|---|---|---|
| PointNet | 83.1 | 79.0 |
| PointNet++ | **84.9** | **82.7** |
| DGCNN | 83.6 | 80.9 |
| NF2vec | 81.3 | 76.9 |
| Functa | 82.8 | 74.8 |
| ENF | 82.2 | 75.4 |

comparably in this task (detailed results and visualizations in Appx. D.6). We again think this attributable to the fact that all shapes are aligned and centered - we further investigate these results in Appx. D.6.1. We additionally include results for point cloud-specific architectures, and NF2Vec - a framework for self-supervised representation learning on 3D data from (non-conditional) NeFs. These results show that ENF only slightly underperforms modality-specific baselines.

**Flood Map Segmentation** For a more challenging segmentation task we apply ENFs on multi-modal flood mapping dataset (Drakonakis et al., 2022). This small dataset (759/85 train/test split) provides dual-modal temporal data; aligned Synthetic Aperture Radar (SAR) and optical satellite images at $256\times256$ resolution obtained by satellites Sentinel 1 and 2 (S1,S2), of disaster sites before and after their flooding, along with corresponding masks that segment the flooded area. The goal is to predict binary segmentation mask given these 4 different input fields. We first train a reconstruction $\mathbf{a}^{\mathrm{SE}(2)}$-ENF $f_{\theta_{\mathrm{recon}}}$ with MAML to obtain a latent $z^{\mathrm{recon}}$ that decodes into the four observations. Next, a segmentation ENF $f_{\theta_{\mathrm{seg}}}$ is trained to predict, given a latent $z^{\mathrm{recon}}$, the binary mask at each location.

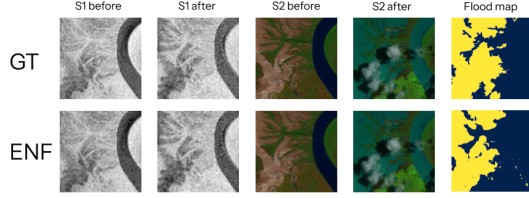

Figure 7: *Top-* OMBRIA test sample of SAR (S1), optical (S2) before and after flooding with ground truth flood map. *Bottom-* ENF reconstructions $f_{\theta_{\mathrm{recon}}}(\cdot; z^{\mathrm{recon}})$ and predicted mask $f_{\theta_{\mathrm{seg}}}(\cdot; z^{\mathrm{recon}})$.

Table 4: Test IOU (↑) for flood map segmentation on OMBRIA, for different observation rates.

| MODEL | PSNR (↑) | IoU (↑) |
|---|---|---|
| | 100% OF $f_{\mathrm{IN}}$ OBSERVED | |
| OmbriaNet | N.A. | 72.36 |
| Functa | 16.77 | 42.75 |
| ENF | **31.65** | **74.00** |
| | 50% OF $f_{\mathrm{IN}}$ OBSERVED | |
| OmbriaNet | N.A. | 27.02 |
| Functa | 16.71 | 42.74 |
| ENF | **31.37** | **73.65** |
| | 10% OF $f_{\mathrm{IN}}$ OBSERVED | |
| OmbriaNet | N.A. | 0.0 |
| Functa | 16.77 | 42.92 |
| ENF | **24.87** | **71.58** |

Table 5: Test IOU (↑) zero-shot resolution transfer on OMBRIA. $f_{\theta_{\mathrm{recon}}}, f_{\theta_{\mathrm{seg}}}$ were trained on $128\times128$ resolution.

| MODEL | PSNR (↑) | IoU (↑) |
|---|---|---|
| | $256\times256$ TEST RESOLUTION | |
| Functa | 16.72 | 37.14 |
| ENF | **28.61** | **72.92** |
| | $128\times128$ TEST RESOLUTION | |
| Functa | 16.71 | 35.48 |
| ENF | **29.31** | **73.21** |
| | $64\times64$ TEST RESOLUTION | |
| Functa | 16.58 | 36.90 |
| ENF | **33.31** | **72.50** |

We evaluate and compare our model against the multi-modal U-Net proposed by Drakonakis et al. (2022). As suggested by Dupont et al. (2022) we trained Functa—unable to fit the training set with MAML—using autodecoding instead. However, we found Functa unable to generalize to test images in this complex low-data setting, collapsing to remembering the training dataset (achieving 31.5 recon PSNR and 93.7 IoU on the $256\times256$ train set). Results (Tab. 4, Fig. 7) show the importance of

Table 6: ERA5 reconstruction $T_t$-MSE↓ and 1-hour forecasting $T_{t+1}$-MSE↓. *MSE between ground truth observations at $T_t$ and $T_{t+1}$.

|  | $T_t$-MSE↓ | $T_{t+1}$-MSE↓ |
|---|---|---|
| Identity* | - | 2.42E-05 |
| Functa | 5.75E-05 | 3.45E-03 |
| ENF | **8.04E-06** | **9.44E-06** |

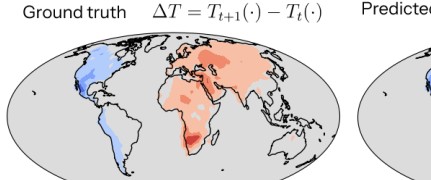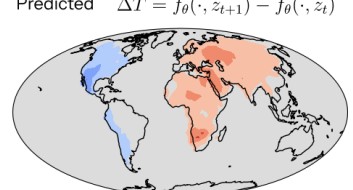

Figure 8: Visualization of ground truth ERA5 test sample and ENF prediction of the change in temperature between observations $T_t$ and $T_{t+1}$. We show the true ($\Delta T$) and predicted ($\Delta \hat{T}$) difference between the temperature maps at $t$ and $t+1$.

inductive biases in complex limited-data regimes. We provide results for *subsampled* observations to simulate missing data, showing the robustness of NeF-based methods to sparsity—ENF performs well even at $10\%$ observation rate where classical convolution-based methods fail. Moreover, we show zero-shot resolution transfer results (Tab. 5) where $f_{\theta_{\text{recon}}}, f_{\theta_{\text{seg}}}$ are trained on $128 \times 128$ resolution data are deployed on $64 \times 64$ and $256 \times 256$ resolution data without fine-tuning, showing the resolution agnostic nature of NeF-representations.

**ERA5 Climate forecasting** Following (Yin et al., 2022; Knigge et al., 2024) we evaluate our NeF-based representation on dynamics forecasting. ERA5 (Hersbach et al., 2019) is a dataset of hourly global temperature observations. We use the dataset as described in Dupont et al. (2021), which contains data defined over $46 \times 90$ latitude-longitude grids. From the training and test sets, we extract 5693 and 443 pairs of subsequent observations $T_t, T_{t+1}$ for train and test sets respectively. Using MAML, we train an ENF $f_\theta$ with bi-invariant $\mathbf{a}^{\emptyset}$ (no symmetries exist in this data) to reconstruct the global temperature state $T_t$ using $z_t$, and optimize a PØNITA MPNN to predict an update $\Delta z_t$ that maps $z_t$ to a latent $\hat{z}_{t+1} = z_t + \Delta z_t$ which decodes into the state at $t+1$, i.e. $T_{t+1} \approx f_\theta(\cdot; z_{t+1}) \approx f_\theta(\cdot; z_t + \Delta z_t)$. Training is done sequentially, i.e. first the backbone $f_\theta$ is optimized and afterward the MPNN is trained, keeping $f_\theta$ fixed. Results (Tab. 6, Fig. 8) show that the latent space of ENF lends itself well for modelling such complex dynamics - where a global latent representation such as Functa seems unable to model the relevant fine-grained details needed for forecasting.

**Image generation** Following Dupont et al. (2022); Bauer et al. (2023), we provide results for diffusion applied to a dataset of latents obtained from pretrained ENFs on CIFAR10 and Celeba$64 \times 64$. As downstream diffusion model, we utilize DiT-B (Peebles & Xie, 2023), a natural choice for our set-latent (training detailed in Appx. C.4). We provide results in FID (Heusel et al., 2017) for

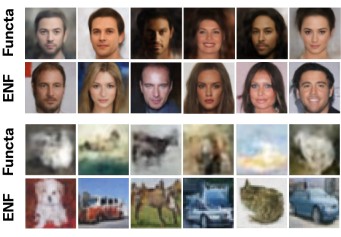

Figure 9: Qualitative samples for generative modelling on CIFAR-10 and Celeba$64 \times 64$.

Table 7: FID for generative modelling on CIFAR-10 and Celeba$64 \times 64$.

| MODEL | CelebA$64 \times 64$ FID ↓ | CIFAR-10 FID ↓ |
|---|---|---|
| GEM | - | 23.8 |
| GASP | 13.5 | - |
| DPF | 13.2 | 15.1 |
| Functa | 40.4 | 78.2 |
| ENF | **33.8** | **23.5** |

unconditional generation in Tab. 7 and samples in Fig. 9. We provide comparison to Functa (Dupont et al., 2022), as well as other frameworks for generative modelling over fields (Du et al., 2021; Dupont et al., 2021; Zhuang et al., 2023)–notably each of these methods is trained on a generative objective and does not support self-supervised pre-training like Functa or ENF. On globally aligned CelebA$64 \times 64$, both Functa and ENF produce perceptually qualitative samples, but unlike ENF, Functa is unable to generalize to CIFAR-10, where data is less homogeneous and not aligned. These results again show clear benefit of a geometrically interpretable latent space for downstream tasks, though previously proposed frameworks specific to generative modelling over fields achieve better performance than ENF. The latter points to a possible area of improvement, and future work could look into incorporating insights from these works into a generative adaptation of the ENF framework, e.g. through latent-space regularization of perceptual consistency as per Du et al. (2021).

## 5 CONCLUSION

Building upon fascinating work using Neural Fields (NeFs) as continuous data surrogates, this paper introduces Equivariant Neural Fields (ENFs); a novel NeF parameterization that re-introduces inductive biases (locality, equivariance) into NeF-based representations. ENF uses a geometry-grounded conditioning variable—a latent attributed point cloud—to achieve an equivariant decoding process, ensuring that transformations in an input field are preserved in the latent space and enabling *steering* of the latent to transform the output signal. This steerability property allows for the accurate representation of geometric information, and for efficient weight-sharing over spatially similar patterns, significantly improving learning efficiency and generalization, as validated on a range of experiments with varying data modalities and objectives.

## 6 REPRODUCIBILITY

All datasets can be downloaded via their cited references without any effort except for the ShapeNet dataset - which requires registration and approval. The pre-processing steps are described in appendix C.3. For all model parameter settings for the ENFs, downstream models or Functa we refer to the appendix C. As supplementary material we added a codebase containing code to reproduce results for the CIFAR10 and OMBRIA experiments. Code for all other experiments will be released during the rebuttal phase of the review process, containing all settings to reproduce the experiments in config files.

## 7 ACKNOWLEDGEMENTS

David Knigge is partially funded by Elekta Oncology Systems AB and a RVO public-private partnership grant (PPS2102). David Wessels is partially funded Ellogon.AI and a public grant of the Dutch Cancer Society (KWF) under subsidy (15059/2022-PPS2). This work used the Dutch national e-infrastructure with the support of the SURF Cooperative using grant no. EINF-9549 and EINF-10544.

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

# A  APPENDIX

## A.1  AUTODECODING AND META-LEARNING

### A.1.1  META-LEARNING

When fitting samples with (Conditional) NeFs using autodecoding (gradient-descent based optimisation of the latent at test-time) (Park et al., 2019), two key challenges emerge: (1) optimising the sample-specific parameters/latent $z$ for a novel sample can be time-consuming -taking e.g. up to 500 gradient updates (Yin et al., 2022)- and (2) more gradient updates to NeF weights may impede downstream performance through a phenomenon known as *overtraining* (Papa et al., 2023) -where the relationship between field $f$ and $z$ is obscured by oversensitivity to high-frequency details. To address the first point, Sitzmann et al. (2020a); Tancik et al. (2021); Dupont et al. (2022) propose a Model-Agnostic Meta-Learning (MAML) (Finn et al., 2017) based optimisation method, enabling the network or latent initialisation to be learned such that each sample can be fitted with just a few gradient steps. More specifically, Dupont et al. (2022) proposes an inner-outer loop structure where the modulations are updated in the inner loop, while the base network weights are updated in the outer loop. This method corresponds to an instance of learning a subset of weights with MAML, also known as Contextual Variable Interaction Analysis (CAVIA) Zintgraf et al. (2019). More recently, Knigge et al. (2024) note that this meta-learning approach also improves downstream performance by imposing structure on the NeF's latent-space. We provide pseudocode for this approach in Alg. 1.

---

**Algorithm 1** Meta-learning ENF

---

Randomly initialize shared base network $f_\theta$
**while** not done **do**
    Sample batch of signals $f$
    Sample random coordinates $\mathbf{x}$
    Initialize latents $z^f \leftarrow \{(p_i, \mathbf{c}_i)\}_{i=1}^N$ for a batch of signals.
    **for all** step $\in 1, ..., N_{\text{inner}}$ and $j \in \mathcal{B}$ **do**
        $z^f \leftarrow z^f - \epsilon \nabla_{z^f} \mathcal{L}_{\text{mse}}\big(f_\theta(x, z^f), f(x)\big)$
    **end for**
    Update ENF: $\theta \leftarrow \theta - \eta \nabla_\theta \mathcal{L}'_{\text{mse}}$
**end while**

---

### A.1.2  AUTODECODING

During our experiments, we found that not all types of signals lend themselves easily to this encoding approach when using ENFs (specifically SDFs and occupancy functions). Although it saves time in inference and adds structure to the latent space, (Dupont et al., 2022) also remark on the limited expressivity of Meta-Learning due to the small number of gradient descent steps used to optimize a latent $z$. As such, for all shape experiments we instead opt for autodecoding (Park et al., 2019), in

which latents and backbone are optimized simultaneously. We provide pseudocode for this approach in Alg. 2.

---

**Algorithm 2** Autodecoding ENF

---

Randomly initialize shared base network $f_\theta$
Initialize latents $z_0^f \leftarrow \{(p_i, \mathbf{c}_i)\}_{i=1}^N$ **for all signals**
**while** not done **do**
    Sample batch of signals $f$
    Sample random coordinates $\mathbf{x}$
    Update latent: $z_{t+1}^f \leftarrow z_t^f - \epsilon \nabla_{z_t^f} \mathcal{L}_{\mathrm{mse}}\big(f_\theta(x, z_t^f), f(x)\big)$
    Update ENF: $\theta_{t+1} \leftarrow \theta_t - \eta \nabla_{\theta_t} \mathcal{L}_{\mathrm{mse}}\big(f_{\theta_t}(x, z_t^f), f(x)\big)$
**end while**

---

### A.1.3 A NOTE ON POSE INITIALIZATION

We noted during our experiments that initialization of the latent poses-i.e. their initial position/orientation in the inner loop-has a significant impact on the reconstruction capacity and stability of the ENF. We found that a good way to initialize the latents is to space them as equidistantly as possible and then adding small Gaussian noise ( $N(0, 1e-3)$), e.g. for 2D images on a perturbed 2D grid. Any orientations are initialized canonically, i.e. all latents are initialized with the same orientation. When defining an equidistantly spaced grid is hard, for example on point clouds or data defined on a sphere, we propose using Farthest Point Sampling on a training grid to initialize positions for the latents.

## B BI-INVARIANT FUNCTION PARAMETERIZATIONS $\mathbf{a}_{m,i}$

The bi-invariants attributes that are used in the experiments section are listed here.

*Translational symmetries* $\mathbb{R}^n$ In this setting, poses correspond to translations $\mathbf{t}_i \in \mathbb{R}^n$:

$$\mathbf{a}_{m,i}^{\mathbb{R}^n} = x_m - \mathbf{t}_i \tag{6}$$

*Roto-translational symmetries* SE(2). In this setting, poses $p_i$ correspond to group elements $g = (\theta_i, \mathbf{t}_i) \in \mathrm{SE}(2)$. We adopt the invariant attribute introduced by (Bekkers et al., 2023):

$$\mathbf{a}_{m,i}^{\mathrm{SE}(2)} = \mathbf{R}_{\theta_i}(x_m - \mathbf{t}_i) \tag{7}$$

*No transformation symmetries*. A simple "bi-invariant" for this setting that preserves all geometric information is given by simply concatenating coordinates $p$ with coordinates $x$:

$$\mathbf{a}_{i,m}^\emptyset = p_i \oplus x_m \tag{8}$$

Parameterizing the cross-attention operation in Eq. 3.1 as function of this bi-invariant results in a framework without any equivariance constraints. We use this in experiments to ablate over equivariance constraints and its impact on performance.

## C EXPERIMENTAL DETAILS

We provide hyperparameters per experiment. We optimize the weights of the neural field $f_\theta$ in all experiments with Adam (Kingma & Ba, 2014) with a learning rate of 1e-4, and an inner step size of 30.0 for $\mathbf{c}_i$ and 1.0 for $p_i$ (increasing inner step size in general speeds up convergence and improves reconstruction - but may also lead to instabilities). For downstream classification we train an equivariant MPNN $F_\psi$, using 3 message passing layers in the architecture specified in Bekkers et al. (2023) conditioned on the same bi-invariant which was used to fit the ENF, with a hidden dimensionality of 256, always trained with learning rate 1e-4. The std parameters $\sigma_q, \sigma_v$ of the RFF embedding functions $\varphi_q, \varphi_v$ are chosen per experiment based on an ablation. In general, increasing both values leads to increased frequency response of the ENF, though generally the model is more susceptible to small change in $\sigma_q$. as well as hidden dim size and the number of attention heads are chosen per experiment, detailed below. We run all experiments on a single H100.

## C.1 IMAGE RECONSTRUCTION, CLASSIFICATION, SEGMENTATION, FORECASTING

**CIFAR10 reconstruction and classification**   For CIFAR10 (Krizhevsky et al., 2009) reconstruction and classification we use a hidden dim of 128 with 3 heads, 25 latents of size 64, a batch size of 32 and restrict the cross-attention operator to k=4 nearest latents for each input coordinate $x$. For $\sigma_q, \sigma_v$ we choose 1.0 and 3.0 respectively. We train the ENF model and the classifier for 100 epochs.

**CelebA64×64**   For CelebA Liu et al. (2015) we use a hidden dim of 256, 36 latents of size 64, a batch size of 2 and restrict the cross-attention operator to k=4 nearest latents for each input coordinate $x$. For $\sigma_q, \sigma_v$ we choose 2.0 and 10.0. We train the model for 30 epochs.

**ImageNet1K reconstruction**   For ImageNet1K (Deng et al., 2009) reconstruction we use a hidden dim of 128 with 3 heads, 169 latents of size 64, a batch size of 2, restricting the cross-attention operator to k=4 nearest latent for each input coordinate $x$. For $\sigma_q, \sigma_v$ we choose 2.0 and 10.0 respectively. We train the model for 2 epochs.

**Ombria**   For OMBRIA Drakonakis et al. (2022) we trained a reconstruction model $f_{\theta_{\text{recon}}}$ with, 256 hidden dim, 4 heads, 169 latents of size 128, a batch size 8. Restricting the cross-attention operator to k=1 nearest latent for each input coordinate $x$. For $\sigma_q, \sigma_v$ we choose 2.0 and 10.0 respectively. The segmentation model $f_{\theta_{\text{seg}}}$ has hidden size of 128, 8 heads, with cross-attention restricted to k=4 nearest latents, trained with batch size 16. For $\sigma_q, \sigma_v$ we choose 2.0 and 3.0 respectively. We train both models, sequentially, for 500 epochs.

**ERA5 forecasting**   For ERA5 forecasting (Hersbach et al., 2019) we train a reconstruction model $f_{\theta_{\text{recon}}}$ with 128 hidden dim, 3 heads, 36 latents of size 64, a batch size of 32. Restricting the cross-attention operator to k=4 nearest latent for each input coordinate $x$. For $\sigma_q, \sigma_v$ we choose 2.0 and 8.0 respectively. Inputs are defined over a latitude longitude $\theta, \phi$ grid, which we map to 3D euclidean coordinates per $\mathbf{x} = [\cos\theta\cos\phi, \cos\theta\sin\phi, \sin\theta]$. We first train the ENF for 800 epochs. As forecasting model, we train a PΘNITA MPNN of 3 layers with 256 hidden dim for 1000 epochs. Both models are trained with a batch size of 32.

As objective, since we don't want to overfit the reconstruction error incurred by fitting a latent $z_t$ to the initial state, we supervise the forecasting model with $L_2$ loss between the decoded output for predicted latent $\hat{z}_{t+1}$ per:

$$L_{\text{forecast}} = ||(f_{\theta_{\text{recon}}}(\cdot; z_t) + \Delta T) - (f_{\theta_{\text{recon}}}(\cdot; \hat{z}_{t+1})||_2^2$$

$\Delta T$ being the ground truth change in temperature, and $\hat{z}_{t+1} = F_{\psi_{\text{forecast}}}(z_t)$.

## C.2 FUNCTA BASELINE MODELS

For the Functa baselines (Dupont et al., 2022), we try to keep as close as possible to the setup defined by the original authors. However, we found that training deeper models in the shape experiments ($\geq$ 8 layers) lead to very unstable training. Instead, for these experiments, we opted to go for shallower models, up to 6 layers. For all experiments we use a hidden dim of 512 and latent modulation size of 512 as used in (Bauer et al., 2023), except for ImageNet1K reconstruction, where we use a 1024 latent modulation. We would like to note here that in all experiments, the Functa baseline has larger parameter count than the ENF models applied to each task (e.g. for CIFAR10, ENF has 522K params where Functa has 2.6M params). Although we did not explore this in-depth, it seems that the proposed ENF representation is more parameter efficient compared to the deep SIREN model defined in (Dupont et al., 2022).

For downstream models we follow (Bauer et al., 2023) and use a 1024 hidden dim 3 layer residual MLP (∼2.1M params). Like in our ENF experiments, we use the same architecture across tasks, only changing the output head to accommodate.

## C.3 SHAPE RECONSTRUCTION AND CLASSIFICATION

**Voxel Dataset and Segmentation**   The voxels are given with the ShapeNet dataset where the segmentation labels are given as point clouds. However, the coordinate frames of the voxels are different, so to align them we mapped both between -1 and 1.

We trained a model $f_{\theta_{recon}}$ with a hidden dim of 128, 3 heads, 27 latents of size 32. We set $\sigma_q, \sigma_v$ for the RFF embedding functions $\phi_q, \phi_v$ to 2 and 10 respectively. For Functa we used, a latent dim of 864 to have the same latent parameters as the conditioning variable used for ENF. As NeF we used a 5-layer Siren with a hidden-dim of 512, $w_0$ is set to 10. The modulation network is a two-layer MLP of hidden sizes 256 and 512.

As is customary for ShapeNet-part segmentation, we condition on the object class and supervise over all segmentation classes with a cross-entropy loss, but only calculate test IoU based on segmentation classes that correspond to the object class. We chose the class-emb dim to be 32 for all settings.

The segmentation NeFs are all trained for 500 epochs with the same parameters as the reconstruction NeF. However, for ENF, we chose $\sigma_q, \sigma_v$ to be 1, 1 for extra stability.

**SDF Dataset**  To create the signed distance functions from ShapeNetCore V2 Chang et al. (2015) objects, we took their meshes and made them water-tight using Point Cloud Utils (Williams, 2022). Afterwards, we sampled a point cloud of 150,000 points from the surface. To create the actual SDF, we perturbed the points with Gaussian noise along the mesh normals, and recalculate the signed distances to the surface. Finally, the dataset consisted of 55 classes with a total number of 42.472 and 5.000 samples for the train and test set respectively.

For ENF we used a latent point cloud of 27 points with context vectors of dimension 32. The std parameters $\sigma_q, \sigma_v$ for the RFF embedding functions $\phi_q, \phi_v$ are 2 and 10 respectively. The hidden dim of the ENF was set to 128 and we used 3 attention-heads. The

For Functa (Dupont et al., 2022) we used a latent modulation of 864 which corresponds to ENFs chosen 27*32 parameters for the conditioning variable. As a NeF we used a 5 layer Siren with an hidden-dim of 512 and a $w_0$ parameter of 15. As a modulation network we used a two-layer MLP with 256,512 hidden-dim.

### C.4 GENERATIVE MODELLING ON ENF LATENT SPACE

For generative modelling experiments CelebA$64\times64$ and CIFAR-10 we train a Diffusion Transformer (DiT-B) (Peebles & Xie, 2023) on ENF latents, utilizing the context vectors as input tokens and their positions as input for an RFF position embedding added to the tokens. We use the same approach for CelebA as for CIFAR10; we train an $\mathbf{a}^{\mathbb{R}^2}$ ENF with MAML on the image dataset, and use this model to obtain sets of "ground truth" latents $z_0 := \{p_i, \mathbf{c}_{i,0}\}_{i=1}^N$ for each image. We then train a DiT-B on a diffusion objective on this latent space, where the forward diffusion kernel is given by:

$$z_t = \{(p_i, \sqrt{\bar{\alpha}_t}\mathbf{c}_{i,0} + \sqrt{1-\bar{\alpha}_t}\epsilon^{\mathbf{c}})\}_{i=1}^N, \tag{9}$$

with $\epsilon^{\mathbf{c}} \in N(0,1)$, i.e. we only add noise to the latent vectors, as we find adding noise to the poses leads to unstable training (something to be investigated in future work). To generate a sample, we take a random set of "ground truth" poses from the training set, and attach a context vector $\mathbf{c}_{i,t} \in N(0,1)$ to each pose to denoise. We supervise the DiT-B with the v objective (Salimans & Ho, 2022). Like (Peebles & Xie, 2023), we use a $t_{max}=1000$ linear noise schedule ranging from $1e-4$ to $1e-2$, and generate samples using DDIM (Song et al., 2020) in 512 steps.

In both settings, we train the diffusion model for 200 epochs using Adam, a constant learning rate of $1e-4$, no weight-decay or dropout, and generate 50k samples to calculate FID.

## D ADDITIONAL RESULTS

### D.1 SIZE OF LATENT POINT-CLOUD

In this section, we delve deeper into the hyper-parameters of the latent point clouds used as conditioning variables. Equivariant Neural Fields can increase the number of parameters used to represent a signal in two ways: by increasing the number of latent points or by expanding the dimensionality of the context vectors. Intuitively, we can either enhance the representational capability of a single region in the input domain or create more, smaller regions with lower representational dimensions.

To gain further insights into how these conditioning variables behave, we train multiple CNFs using different latent point-cloud configurations. In these experiments, the chosen latent dimension or number of latent points is adjusted to keep the total number of parameters as close as possible across configurations. We trained each model for 400 epochs, as only small improvements occurred beyond this point and the overall trend was already clear. After fitting the ENF, we used the meta-learned representation to perform classification tasks. We employed a simple message-passing GNN, which we trained for 20 epochs, after which performance improvements began to degrade. Below, we present the ablation results for these different approaches to increasing representational capabilities.

Table 8: Reconstruction PSNR (db↑) and ACC (%↑) on CIFAR10 for different parametrisations of the latent point-clouds, i.e. varying $N, d$ in $z := \{\mathbf{c}_i \in \mathbb{R}^d\}_{i=1}^N$.

| # LATENTS(N) | LATENT DIM(D) | # PARAMS | PSNR | ACC (%) |
|---|---|---|---|---|
| 1 | 1600 | 1600 | 22.69 | 53.21 |
| 4 | 400 | 1600 | 29.14 | 64.98 |
| 9 | 178 | 1602 | 35.49 | 73.54 |
| 16 | 100 | 1600 | 39.93 | 77.09 |

## D.2 ABLATION ON GAUSSIAN SPATIAL WINDOWING AND kNN APPROXIMATION

To evaluate the impact of Gaussian spatial windowing (GSW) and the k-Nearest Neighbors (kNN) approximation in the proposed method, we trained four models on the CIFAR10 dataset: one with both features disabled, one with only kNN enabled, one with only GSW enabled, and one with both enabled. Besides evaluating the difference in reconstruction capabilities, we are mainly interested in the downstream performance. We argue that the introduced locality enhances latent-space structure by improving weight-sharing across local-patches.

After training the models, we used the different ENF models to generate latent representations for CIFAR10 classification. The results are shown in Table 9. While reconstruction performance remains almost consistent across the different setups, GWS significantly improves downstream classification accuracy. Moreover, the kNN approximation does not negatively affect either reconstruction nor classification performance. Interestingly, kNN even provides a slight improvement even without GWS. We hypothesize that this improvement comes from kNN introducing an implicit form of windowing—not by modifying attention values directly but by limiting the set of attention values considered. To conclude, there can be observed that the introduced locality in CNF latents does improve the downstream performance.

Table 9: Reconstruction PSNR (db↑) and ACC (%↑) on CIFAR10 to ablate the Gaussian spatial windowing and kNN approximation.

| ENF SETUPS | PSNR (DB ↑) | ACC (% ↑) |
|---|---|---|
| ENF | 39.1 | 70.1 |
| ENF + kNN | 40.8 | 72.8 |
| ENF + GWS | TBD | TBD |
| ENF + GWS + kNN | 42.2 | 82.1 |

## D.3 TRANSFORMING THE LATENT POINT-CLOUD

We provide visualizations for transformations applied to the latent pointclouds for different bi-invariants $\mathbf{a}$ in Fig. 10.

## D.4 ENF WITH GEOMETRY-FREE LATENT SETS

To further investigate what design choices the performance of ENF results from, we provide an ablation on CIFAR10 over a geometry-free implementation of ENF. We do this by removing the pose information from the latent set, i.e. we set $z := \{\mathbf{c}_i\}_{i=1}^N$, and use a "bi-invariant" that is only a function of $x_j$, $\mathbf{a}_{i,j} = x_j$. Since now latents do not have a position, we remove the Gaussian windowing and KNN approximation, but keep the rest of the ENF architecture as well as the hyper-parameters used identical to the settings reported in Appx. C under 'CIFAR10 reconstruction and

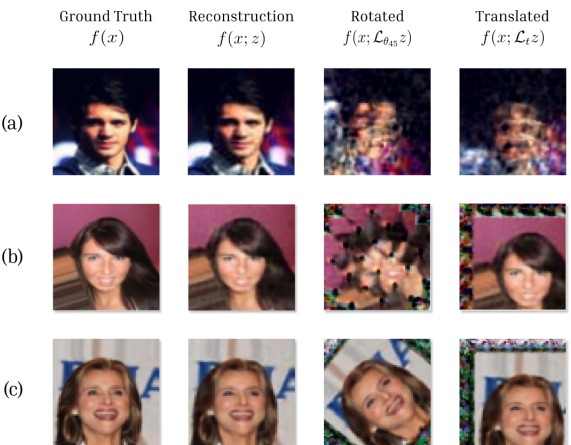

Figure 10: Transformations applied to latents $z$ for different bi-invariants $\mathbf{a}$. (a) $\mathbf{a}_{m,i}^{\emptyset}$ is not bi-invariant to any transformations, (b) $\mathbf{a}_{m,i}^{\mathbb{R}^2}$ is bi-invariant to translations, producing distorted patterns on rotation and (c) $\mathbf{a}_{m,i}^{\mathrm{SE}(2)}$ is bi-invariant to roto-translations; the output $f_\theta(x;z)$ equivaries with both rotations and translations applied to $z$.

classification'. We observe highly unstable training during the reconstruction phase, and reconstruction performance on the test set converges to 22.3. We think this attributable to the fact that now any update to one of the latent codes affects the output of the NEF globally, leading to a much more complex optimization landscape. This highlights another advantage of either having a single global latent, or using locality as inductive bias; optimization of single or locally responsible latents seems to lead to a simpler optimization landscape compared to optimizing a set of global latents.

We subsequently train a simple 4 layer transformer with hidden dim 256 and 4 heads as classifier. Note that this transformer uses no positional encoding, since the latent $z$ in this setting has no associated geometry/positional information. We train for 500 epochs on the augmented dataset, after which training accuracy has converged to 95%. We observed overfitting early into training. Utilizing early stopping, best performance was achieved after just 5 epochs, yielding a test set accuracy of 0.47. These observations (Tab. 10) are in line with the outcome of our other experiments; geometry-grounded latents are more informative for downstream tasks.

Table 10: Reconstruction PSNR (db↑) and classification test accuracy (%↑) on CIFAR10 when ablating over latent geometry.

|  | PSNR | ACC (%) |
|---|---|---|
| Functa | 38.1 | 68.3 |
| ENF w/ pose-free latents | 22.3 | 47.9 |
| ENF w/ $\mathbb{R}^2$ latents | **42.2** | **82.1** |

### D.5 DETAILS ON COMPUTATIONAL EFFICIENCY

To allow for more fine-grained comparison of our method with previous work, we provide details on time and memory efficiency of our approach on the CIFAR10 classification experiments listed in Tab. 1 with $\mathbf{a}^{\mathbb{R}^2}$, when both Functa and ENF are fit using MAML with 3 inner loop steps. Moreover, we compare efficiency also when ablating over the KNN approximation of the attention operation. We report estimated FLOPs (obtained through JAX's AOT api), GPU memory usage per sample and training time per epoch for a batch size of 32. We see (Tab. 11) that the naive implementation that does not truncate the attention operator is significantly more FLOP-intensive and memory intensive compared to Functa (Dupont et al., 2022) and the KNN approximate implementation. Functa in all settings does have considerably higher runtime, attributable to its relatively deep sequential architecture compared to the shallow single layer architecture of ENF. These results show that, besides

being more performant on fine-grained downstream tasks, ENF also scales favourably compared to Functa.

Table 11: Computational efficiency of ENF with and without KNN approximation to the Functa baseline for the CIFAR10 experiment.

| | FLOPS ($\times 10^9$) | GPU MEMORY (GB/SAMPLE) | TIME PER EPOCH (S) |
|---|---|---|---|
| Functa | 28.3 | 0.61 | 2864 |
| ENF w/o KNN approx. | 104.5 | 1.83 | 1801 |
| ENF w KNN approx. | 22.7 | 0.40 | 207 |

### D.6 ADDITIONAL SHAPENET-PART SEGMENTATION RESULTS

Below we show the full table with ShapeNet-Part segmentation results with IoUs per class in table 12 and some qualitative examples in figure 11.

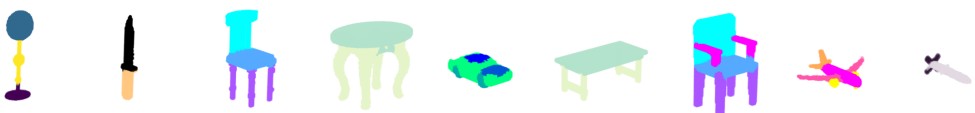

Figure 11: Qualitative examples drawn randomly from the ShapeNet segmentation test set.

Table 12: Segmentation class and instance averaged IOU (↑) on ShapeNet, and mIoUs per class.

| MODEL | INST MIOU | CLS MIOU | AIRPLANE | BAG | CAP | CAR | CHAIR | EARPHONE | GUITAR | KNIFE | LAMP | LAPTOP | MOTORBIKE | MUG | PISTOL | ROCKET | SKATEBOARD | TABLE |
|---|---|---|---|---|---|---|---|---|---|---|---|---|---|---|---|---|---|---|
| PointNet | 83.1 | 79.0 | 81.3 | 76.9 | 79.6 | 71.4 | 89.4 | 67.0 | 91.2 | 80.5 | 80.0 | 95.1 | 66.3 | 91.3 | 80.6 | 57.8 | 73.6 | 81.5 |
| PointNet++ | **84.9** | **82.7** | **82.2** | **88.8** | **84.0** | **76.0** | **90.4** | 80.6 | **91.8** | 84.9 | **84.4** | 94.9 | **72.2** | **94.7** | **81.3** | 61.1 | 74.1 | 82.3 |
| DGCNN | 83.6 | 80.9 | 80.7 | 84.3 | 82.8 | 74.8 | 89.0 | **81.2** | 90.1 | **86.4** | 84.0 | **95.4** | 59.3 | 92.8 | 77.8 | **62.5** | 71.6 | 81.1 |
| NF2vec | 81.3 | _76.9_ | 80.2 | 76.2 | _70.3_ | 70.1 | 88.0 | _65.0_ | _90.6_ | 82.1 | 77.4 | 94.4 | 61.4 | _92.7_ | _79.0_ | _56.2_ | 68.6 | 78.5 |
| Functa | _82.8_ | 74.8 | _82.1_ | 72.5 | 40.2 | 72.5 | _88.5_ | 60.9 | 89.2 | _82.2_ | 80.1 | _93.9_ | _63.8_ | 90.6 | 77.3 | 46.3 | _76.0_ | _81.5_ |
| ENF | 82.2 | 75.4 | 80.7 | _77.2_ | 42.1 | _73.2_ | 87.7 | 64.4 | 89.4 | 79.6 | _80.6_ | 93.8 | 62.7 | 91.8 | 76.9 | 52.5 | 74.1 | 80.6 |

### D.6.1 SHAPENET-PART SEGMENTATION WITHOUT SHAPE INFORMATION

Further investigating the results obtained in the ShapeNet Part classification task, we train an ENF $f_{\theta_{\text{seg}}}$ without conditioning on $z^{\text{recon}}$ - i.e. without any shape-specific conditioning but instead only conditioning on the object class. This model obtains class and instance mIoU of 64.3 and 69.2 respectively, indicating that a lot of points in this dataset can be correctly segmented purely based on their absolute position, and as such the backbone NeF model does not need to capture to perform decently on this dataset - though we would expect additional geometric to help with performance.

