# OpenReview forum: "Grounding Continuous Representations in Geometry: Equivariant Neural Fields"
_ICLR.cc/2025/Conference — ICLR 2025 Poster_

### Official Review · Reviewer_9X4V · 2024-10-30

**Soundness:** 2
**Presentation:** 3
**Contribution:** 2
**Rating:** 6
**Confidence:** 4

**Summary:**

The paper presents a method for conditioning a neural field using a set of $SE(n)$ equivariant local latents. The aim is to enhance downstream task performance by operating on the neural field’s learned latent representation, rather than on discrete samples from the continuous signal as in conventional approaches. It outlines the necessary conditions for an equivariant latent representation in neural fields and adapts a cross-attention architecture to support these conditions. The approach is evaluated across a wide variety of tasks.

**Strengths:**

The paper is well-written and organized, with a clearly defined method supported by formal definitions.

The proposed solution is simple and intuitive for enhancing CNFs with local equivariant features.

I appreciate the variety of dataset types used in the experiments.

**Weaknesses:**

Lack of motivation for using CNF latent encodings in downstream tasks

The paper does not explicitly discuss the motivation for using latent encodings of CNFs for downstream tasks. It seems that one advantage might be the ability to utilize more data for training since the reconstruction training stage does not require labeled data. This raises a follow-up question:
What benefit do latent features learned through continuous reconstruction (decoder) have over latent features learned through reconstructing a discrete sample?  It seems that a continuous decoder could enable the learning of discretization-agnostic features.  Is there another motivation for using latent encodings of CNFs for downstream tasks?  The purpose behind their use in enhancing downstream tasks remains somewhat unclear.

The motivation for using local CNF latent encodings could be framed more clearly.

The paper states that a notable limitation for conventional CNF (ln 51): “each field is encoded by a global variable”. However, this statement about cnfs limitations seems to be partially accurate. In fact, this approach for latent space modeling also has some clear advantages. For example, interpolating between two latents to generate novel signals is far more natural with a global latent structure, whereas a local latent structure requires solving the complex problem of finding correspondences between latent points. Thus, the characterization of a tradeoff rather than a limitation may be more appropriate.
Additionally, to address the limitations of a global latent, why not employ an encoder-decoder architecture with gradually decreasing spatial dependency in the latent representation (similar to a UNet)? This approach would provide a final latent that incorporates both local and global information. The rationale for restricting the model to an auto-decoder-style architecture remains unclear.

Overclaiming on Geometry-Appearance Separation in Neural Fields

The paper claims that the proposed method “separates geometry from appearance” in its representation. My understanding is that this refers to the structure of pose-appearance tuples in the latent space. However, how does the method ensure that only appearance information is captured in  $c_i$ ? This seems to rely solely on  $c_i$  being an $SE(n)$ invariant feature. Yet, some relevant geometric features are also invariant (e.g., shape volume), while some equivariant features can relate to appearance (e.g., how an object’s appearance changes are affected by material reflectance features under rotation). Consequently, enforcing a latent structure of invariant and equivariant features may not be sufficient to achieve true separation. Is there empirical evidence to support the above claim about separation?

Unclear reconstruction results

The paper claims: “Results show that ours as well as the baseline models struggle with accurately reconstructing the underlying shape from the SDF point clouds”. Given the inaccuracies in reconstruction, how can the learned features be effectively used for downstream tasks? Additionally, it’s unclear why this model underperforms in reconstruction compared to [3]. Both architectures appear similar (apart from the equivariant features), yet [3] reports more accurate reconstruction results.

Unclear segmentation results

The choice of ShapeNet as the dataset for segmentation evaluation is questionable, as it is an aligned dataset (line 468). A better alternative might be to use non-aligned datasets, such as those used for human-body segmentation in [4] and [5]. Another option would be to unalign ShapeNet by applying a random $SE(3)$ transformation to each data point. Additionally, it’s unclear if the point cloud-specific architectures were also trained with a reconstruction pretext stage.

Additional comments.

Figure 8 is uninformative on its own without comparison to other methods, showcasing some of the proposed method qualitative benefits/limitations.

Conditioning with k-nearest neighbors appears to restrict the smoothness of the modeled field to be at most continuous, while the data signals are at least differentiable.

The Steerability property for CNFs has also been defined and utilized in prior works, such as [1] and [2].

[1] Frame Averaging for Equivariant Shape Space Learning. Matan Atzmon, Koki Nagano, Sanja Fidler, Sameh Khamis, Yaron Lipman.

[2] Vector Neurons: A General Framework for SO(3)-Equivariant Networks. Congyue Deng, Or Litany, Yueqi Duan, Adrien Poulenard, Andrea Tagliasacchi, Leonidas Guibas.

[3] Biao Zhang, Jiapeng Tang, Matthias Niessner, and Peter Wonka. 3dshape2vecset: A 3d shape representation for neural fields and generative diffusion models.

[4] Approximately Piecewise E(3) Equivariant Point Networks. Matan Atzmon, Jiahui Huang, Francis Williams, Or Litany.

[5] Generalizing neural human fitting to unseen poses with articulated se (3) equivariance. Haiwen Feng, Peter Kulits, Shichen Liu, Michael J Black, and Victoria Fernandez Abrevaya

**Questions:**

I would appreciate a response regarding the weaknesses and questions mentioned above.

---

> ### Author Response · Authors · 2024-11-22
> **Initial response to reviewer 9X4V**
>
> We thank the reviewer for their thorough assessment of our manuscript and his appreciation of the simple and intuitive proposed solution. Moreover, we appreciate the thoughtful questions and points of discussion raised by the reviewer, which we feel will strengthen the work. Below we will elaborate on the questions and address the weaknesses.
>
> **Lack of motivation for using CNF latent encodings in downstream tasks** The reviewer notes that we do not explicitly discuss the motivation for using Neural Field (NeF) based representations. Building on an increasingly large body of work utilizing Neural Fields as representations for a host of tasks (see e.g. [11] for a broad overview of use-cases in 2D and 3D reconstruction, generative modeling, compression, robotics, forecasting), we indeed realize we left this motivation mostly implicit in the current version of the manuscript. We agree with the reviewer that explicitly adding this motivation strengthens the manuscript, and dedicate some space to this in the introduction of the revision we attach to this response (ln 040 -> 044). We agree with the advantages stated by the reviewer and will further elaborate below.
>
> The first advantage of NeF-based representations results from its discretization/resolution agnostic nature; a NeF-representation is not tied to a grid and as such is able to seamlessly transfer over on different discretizations / samplings of the same underlying data, a principle corroborated by the findings in our experiments on zero-shot resolution transfer and robustness to sparsity on the OMBRIA dataset.
>
> Another advantage of NeF-based representations is that they are applicable to a range of spatial data modalities and geometries; as long as there is coordinate-signal data available, it is possible to fit a NeF-based representation to this data. As a result, this unifies models applicable to these different modalities and geometries that classically require their own specific engineering efforts. We show this in our experiments; we use the same downstream architecture for forecasting over spherical data as we use to classify image data. We feel this transferability is a desirable property compared to designing specific architectures for tasks and data types as is the case for classical grid-based or grid-free (point cloud) representations, since this in turn allows for the transfer of modeling principles between modalities and geometries.
>
> A third, somewhat adjacent, advantage is that these continuous representations scale better by increasing resolution size; NeF-based representations are shown to scale with signal complexity rather than discretization resolution, which is shown in figure 2 of [3].
>
> For these reasons, we feel the pursuit of NeF-based continuous signal representations is worthwhile. We amend our introduction to better reflect this.
>
> **Lack of motivation for using local CNF latent encodings**
> The reviewer points out that we are harsh in our description of the use of global latents in e.g. Functa, denoting them as a limitation. We would like to point out that we do not necessarily oppose the use of global latents, or find them inherently limiting, only that through the use of a single global latent no explicit geometric information (e.g. position, orientation, relative position) on features in the signal is retained to be leveraged by downstream models, but instead these features are necessarily represented implicitly, limiting performance (as shown in our experiments) on tasks that require fine-grained reasoning (classification, segmentation). The reviewer highlights that there is an argument to be made for certain cases where this global latent is desirable, e.g. the notion of latent-space interpolation would be quite complex with our proposed local set-latent, but is very natural when representing a signal with a single latent. In settings with globally aligned data, this is naturally true. We do see the need for a nuanced representation of our method in contrast to previous works, and rewrote the intro section (ln 074 -> 079) to indicate the specific sort of tasks we think are limited by implicit representation of geometric information, and indicate the tradeoff between ease of downstream use (global latents) and performance (equivariant local latents).

---

> > ### Author Response · Authors · 2024-11-22
> > **Continuation of first response to reviewer 9X4V**
> >
> > Additionally, we do agree that an UNet-like decoder-only variation of our ENFs would still be able to incorporate local and global information in a single vector representation. This is an interesting direction for future work, but we consider this a significant deviation from our proposed solution and outside the paper’s scope. It would require working with multiple sets of latents (one set per scale) or organizing the latent space in a hierarchical fashion which is not straight forward. We do consider this a novel and potentially high impact direction which we leave for further research. We further note that a classical u-net type encoder-decoder approach would not be able to learn compressed representations since due to the skip connections the reconstruction loss is trivial. Only when skip connections are removed we obtain a bottle neck that allows for representation learning, but then it is just an auto-encoder. It is precisely the decoder-only approach that allows for discretization-free representation learning, following the Functa paradigm [3].
> >
> > **Overclaiming on Geometry-Appearance Separation in Neural Fields**
> > In line 086 we write that we propose “representations that separate geometry from appearance”. The reviewer argues that this claim is a bit too strong, as we can give no theoretic guarantees showing that by modelling $\mathbf{c}_i$ through group-invariant features we separate out all geometric information. This is a valid remark, as such we nuance this claim by amending this part of the introduction (ln 092) to be more specific on what signal attributes the proposed representation separates. ENF representations explicitly encode for geometric information in the poses and their relative positioning, that is, it separates out the pose (e.g. location, orientation) and (SE(n)-)invariant appearance of features in the signal. This is a capability that is absent in other methods in neural field literature. Our results for downstream tasks very clearly show the benefit of this in fine-grained tasks, and we argue these results are attributable at least in part exactly due to this separation.
> >
> > Note that in its current form, ENF indeed only supports scalar output fields, i.e. feature quantities that are invariant under transformation. A very interesting extension alluded to by the reviewer would be to generalize ENF to support modelling of fields of higher order features, e.g. vector fields, that transform equivariantly. This could find applications in many physics problems such as PDE modelling. A possible way to approach this could be through the framework of Clifford group equivariance [15], which naturally supports modelling of higher order features (e.g. vector fields) in neural networks.
> >
> > **Unclear reconstruction results**
> > The reviewer asks how the latent-representations of inaccurately reconstructed samples are still effectively used for downstream tasks. Interestingly, previous work by [4] shows that reconstruction performance for Neural Field representations is not indicative of performance as downstream representation in classification tasks. In fact, it seems that underfitting helps downstream performance to some extent, which the authors attribute to divergence of parameter-space representations for NeF-representations fit with the higher number of SGD steps required to get better reconstruction. However, we agree that many downstream tasks indeed require good reconstruction performance for the use of NeF-representations, e.g. performance of generative modelling in NeF latent spaces is constrained by the ability of this latent space to represent the original data distribution in the first place, and so downstream performance in such tasks on poorly fit NeF latents will always be limited. We intended for these results to show the applicability of ENF to different representations of the same data modality (occupancy vs. SDF), but see how these results might confuse the reader. As explained below, we choose to de-emphasize the experiments on ShapeNet and instead follow [3] in providing additional experimental results for generative modelling over ENF latents using diffusion.

---

> ### Author Response · Authors · 2024-11-22
> **Continuation of first response to reviewer 9X4V**
>
> Regarding 3DShape2Vecset [5], we indeed observed a difference between reconstruction performance on occupancy of shapes (ENF obtains 0.929 IOU as measured over the largest 16 classes, where 3DShape2Vecset obtains 0.967 IOU measured over the largest 7 classes). We feel this can at least partially be explained by the difference in latent and model size; 3DShape2Vecset uses a set of 512 latents of dimension 512 (512^2 parameters) to represent a single shape, whereas our model as well as the Functa baseline only has ~800 degrees of freedom. We were unable to reproduce these results from the available code however, and since we’re approaching NeF representations for data representations more broadly – not focussing on shape data specifically –, we left this model out of our comparison. In our response to reviewer 4oVU, we do show downstream results when ablating over the pose information in our latent set parameterization, i.e. when using a set of pose-free latents as conditioning variable–a parameterization comparable to generalizing 3DShape2Vecset to arbitrary signal data. Results for CIFAR classification show the clear benefit of having a geometrically grounded localized latent set (82.1% -> 47.9% accuracy).
>
> **Unclear segmentation results**
> We acknowledge the reviewer's concern that the choice of segmentation on the Shapenet-part dataset is questionable due to the global alignment of the dataset. We do think that these results, showing performance on-par with non-equivariant NeF-based baseline methods, allow for an assessment of performance of our method in the setting when the dataset does not exhibit global symmetries, i.e. it shows ENF performs on par also with these equivariance constraints. Comparing this setting with more classical point cloud specific methods shows that modality agnostic NeF-based representations only perform marginally worse. We investigate these results more in-depth in Appx. D, but since other reviewers also identified the segmentation results as a possible point of confusion for the reader, we believe that replacing this experiment in the main text with a generative modeling experiment will more effectively highlight the advantages of our method. We modify the method section by moving Fig. 8 to the appendix and summarizing the segmentation results more concisely in the main body.
>
> The reviewer also pointed out that it is unclear whether the point-cloud methods used reconstruction as a pretext phase, this is not the case, the results for point cloud specific methods as reported in [6] are trained only on the segmentation task.
>
> **Additional comments**
> - The reviewer argues that figure 8 is uninformative without any comparison to other methods, we do agree with this and will move the figure to the appendix.
>
> - The reviewer argues that our k-nearest neighbors approach to make the attention operation more efficient would restrict the smoothness of the resulting function. In practice, we find no training instabilities resulting from our parameterization, we think attributable to our use of the Gaussian window. The Gaussian window forces the attention coefficients to zero as distance grows and hence gradients for such latents naturally vanish, making the KNN approximation functionally equivalent.
>
> - The reviewer argues that the steerability property for CNFs has also been defined in [7, 8]. We thank the reviewer for bringing these papers to our attention and we will refer to them in the main text. Especially [7] provides an interesting perspective on the notion of steerability in shape spaces. Though similar, we argue that our viewpoint on steerability in CNFs through bi-invariance constraints on the binary function that parameterizes the CNF is a useful and novel one, in that it allows for simple equivariant NF implementations. However, steerability is indeed a widely used concept in deep learning e.g. [7, 8, 12, 13, 14]. In fact, the definition of our steerability constraint via bi-invariants in Lemma 1 of our manuscript is similar to proofs in [9, 10] which show that 2 argument kernels should be bi-invariant to be equivariant. After analyzing our manuscript again we do agree that in these sections references to the denoted works should be added, and we do so in the revision we attach to this response.

---

> > ### Author Response · Authors · 2024-11-22
> > **Final part of response to reviewer 9X4V**
> >
> > We thank the reviewer for their clear effort and thoughtfulness in reviewing our manuscript. We feel addressing the concerns noted by the reviewer strengthened our manuscript considerably. We’re happy to continue the discussion, please let us know if any points of concern remain.
> >
> > [1] Mildenhall, B., Srinivasan, P. P., Tancik, M., Barron, J. T., Ramamoorthi, R., & Ng, R. (2021). Nerf: Representing scenes as neural radiance fields for view synthesis. Communications of the ACM, 65(1), 99-106.
> >
> > [2] Yin, Y., Kirchmeyer, M., Franceschi, J. Y., Rakotomamonjy, A., & Gallinari, P. (2022). Continuous pde dynamics forecasting with implicit neural representations. arXiv preprint arXiv:2209.14855.
> >
> > [3] Dupont, E., Kim, H., Eslami, S. M., Rezende, D., & Rosenbaum, D. (2022). From data to functa: Your data point is a function and you can treat it like one. arXiv preprint arXiv:2201.12204.
> >
> > [4] Papa, S., Valperga, R., Knigge, D., Kofinas, M., Lippe, P., Sonke, J. J., & Gavves, E. (2024). How to Train Neural Field Representations: A Comprehensive Study and Benchmark. In Proceedings of the IEEE/CVF Conference on Computer Vision and Pattern Recognition (pp. 22616-22625).
> >
> > [5] Biao Zhang, Jiapeng Tang, Matthias Niessner, and Peter Wonka. 3dshape2vecset: A 3d shape representation for neural fields and generative diffusion models
> >
> > [6] De Luigi, L., Cardace, A., Spezialetti, R., Ramirez, P. Z., Salti, S., & Di Stefano, L. (2023). Deep learning on implicit neural representations of shapes. arXiv preprint arXiv:2302.05438.
> >
> > [7] Frame Averaging for Equivariant Shape Space Learning. Matan Atzmon, Koki Nagano, Sanja Fidler, Sameh Khamis, Yaron Lipman.
> >
> > [8] Vector Neurons: A General Framework for SO(3)-Equivariant Networks. Congyue Deng, Or Litany, Yueqi Duan, Adrien Poulenard, Andrea Tagliasacchi, Leonidas Guibas.
> >
> > [9] Cohen, T. S., Geiger, M., & Weiler, M. (2019). A general theory of equivariant cnns on homogeneous spaces. Advances in neural information processing systems, 32.
> >
> > [10] Bekkers, E. J. (2019). B-spline cnns on lie groups. arXiv preprint arXiv:1909.12057.
> >
> > [11] Xie, Y., Takikawa, T., Saito, S., Litany, O., Yan, S., Khan, N., ... & Sridhar, S. (2022, May). Neural fields in visual computing and beyond. In Computer Graphics Forum (Vol. 41, No. 2, pp. 641-676).
> >
> > [12] Brandstetter, J., Hesselink, R., van der Pol, E., Bekkers, E. J., & Welling, M. (2021). Geometric and physical quantities improve e (3) equivariant message passing. arXiv preprint arXiv:2110.02905.
> >
> > [13] Cesa, G., Lang, L., & Weiler, M. (2022). A program to build E (N)-equivariant steerable CNNs. In International conference on learning representations.
> >
> > [14] Cohen, T. S., & Welling, M. (2016). Steerable cnns. arXiv preprint arXiv:1612.08498.
> >
> > [15] Ruhe, D., Brandstetter, J., & Forré, P. (2024). Clifford group equivariant neural networks. Advances in Neural Information Processing Systems, 36.

---

> > > ### Comment · Reviewer_9X4V · 2024-11-25
> > >
> > > Thank you to the authors for providing a detailed rebuttal. While some of my concerns have been adequately addressed, some issues remain.
> > >
> > > The motivation for using CNF latent encodings in downstream tasks remains unclear. Based on the revised advantages, here are my remaining concerns:
> > >
> > > Resolution-Agnostic Benefit: There does not appear to be sufficient evidence supporting this claimed advantage. For example, a simple baseline could involve upsampling or downsampling the input and output before or after applying a CNN in the zero-shot experiment. Additionally, it would be valuable and necessary to test this benefit on a larger-scale dataset than those used in Tables 4 and 5, as the current datasets are notably small.
> > >
> > > Transferability vs. Task-Specific Architectures: While transferability is indeed a desirable property compared to designing task-specific architectures, such as those tailored to grid-based or grid-free (e.g., point cloud) representations, it should be noted that task-specific architectures also have significant advantages. For example, the development of CNNs specifically optimized for spatial data significantly outperformed MLPs in certain contexts. Claiming transferability as an outright advantage may therefore be an overstatement, as it overlooks the practical benefits of architectures designed to exploit the unique properties of specific data types.
> > >
> > > “This is a capability that is absent in other methods in neural field literature.” Could the authors clarify what specific capability they are referring to? If it pertains to conditioning a neural field (NeF) on invariant and equivariant features, I find this statement confusing, particularly in light of the authors’ comments regarding the works of [7] and [8].
> > >
> > > Regarding the segmentation results, I believe the work would benefit from an evaluation on a non-aligned, large-scale dataset. I encourage the authors to consider this in future work.

---

> > > > ### Author Response · Authors · 2024-11-26
> > > >
> > > > We thank the reviewer for the engaging comments and questions, and are happy to see we were able to clear up some of the reviewer’s concerns! We hope to continue the conversation with this response, and are eager to hear your thoughts.
> > > >
> > > > **Resolution agnostic benefit** The reviewer raises concerns regarding the benefit of resolution agnosticity in NeFs and how we approach showing this empirically. However, we would like to stress that we do not intend to position resolution agnosticity of ENF as a benefit in itself, but try to convey that the discretization-free nature of NeFs, and their resulting resolution agnosticity, is beneficial in real-world data settings.
> > > >
> > > > To clarify, we see zero-shot resolution transfer (resolution agnosticity) as an expression of the inherent discretization-free nature of NeF-based methods; ability to inherently handle sparse observations and changing test-time grids are two examples of benefits of grid-free representations, which is why we chose to group the experiments in Tab. 4 and 5. We appreciate the reviewer’s suggestion to include a comparison with classical CNNs using upsampling and downsampling in the zero-shot resolution transfer experiment (Tab. 5). While such a comparison is feasible and could demonstrate the how CNNs operate at different resolutions with pre-/post-processing (even though at full resolution ENF already outperforms the U-Net based baseline), it would miss the core aim of this section of our experiments: to showcase the discretization-free nature of NeF-based representations, i.e. their ability to handle sparse and irregular grids – as well as arbitrary test-time changes to the observation grids, which is inherently beyond the scope of grid-dependent CNN architectures.
> > > >
> > > > NeFs like ENF are fundamentally designed to operate directly on irregular, sparse, or non-grid-aligned data. This is a critical distinction, as conventional CNNs, even with upsampling or downsampling, rely on a regular grid structure to function. To illustrate this distinction, consider the experiments in Table 4, where ENFs demonstrate robustness in handling sparse observations (e.g., 10% of input data observed). In these cases, as we empirically show, it is not feasible to apply CNN-based methods, as the lack of a complete grid renders the U-Net baseline architecture ineffective. Similarly, in Table 5, ENFs seamlessly generalize across resolutions in a zero-shot setting due to their resolution-agnostic formulation, which eliminates the need for explicit pre- or post-processing steps like upsampling or downsampling.
> > > >
> > > > By combining these results, we highlight a broader advantage of ENFs: their flexibility to operate on both dense and sparse data without requiring the architectural modifications or pre-/post-processing necessary for grid-based models. This capability reflects their resolution-agnostic and grid-free design principles, as well as their potential to unify tasks across varying spatial domains.
> > > > We’re very interested to hear if the reviewer is able to align with this reasoning, and welcome their thoughts.
> > > >
> > > > We add a clarification in the manuscript regarding the goal of the experiments on the OMBRIA dataset to elaborate. This adjustment emphasizes that our objective is to showcase a paradigm of representation that is fundamentally different from classical architectures, providing a robust foundation for grid-free and multi-resolution tasks.
> > > >
> > > > Regarding the size of the OMBRIA dataset; although experiments on larger-scale datasets would give valuable insights on scaling performance of our method, we respectfully disagree with the reviewer that it is necessary to evaluate on larger-scale datasets to demonstrate the discretization-free capabilities of ENFs. The OMBRIA dataset, used in our experiments (Tables 4 and 5), represents a real-world scientific dataset where data scarcity is an inherent challenge. Labelling such data requires expertise, making it expensive, and hence the ability for DL-based methods to generalize well even in these low-data regimes is vital for successful application. In these settings, handling data sparsity becomes increasingly important, as exactly the ability to handle noisy observations (i.e. sparse observations) and data over different resolutions allows the model to learn from–and be applied to–a larger set of data points.

---

> > > > > ### Author Response · Authors · 2024-11-26
> > > > >
> > > > > **Transferability and task-specific architectures** We are glad to see the reviewer agrees that transferability is a desirable property in model design. We would like to clarify that we are not opposed to the inclusion of datatype-/ task-specific considerations in model design. In fact, we show the benefits of using such task-specific inductive biases throughout our experiments in the form of locality/equivariance constraints.
> > > > >
> > > > > The reviewer cites the success of the CNN architecture as an example of task-specific design optimized for spatial data. We respectfully argue that this example supports our perspective; Classical CNNs were indeed designed with task-specific considerations, namely leveraging locality and weight-sharing as inductive biases for spatially structured data. Consider e.g. the following passage taken from LeCun et al. (1989); “We have required our network to do this by constraining the connections in the first few layers to be local. In addition, if a feature detector is useful on one part of the image, it is likely to be useful on other parts of the image as well.”
> > > > >
> > > > > The classical implementation of a CNN has been designed specifically for regularly gridded data through its use of a discrete set of weights to identify kernel values. This is simply an implementation of the notions of locality and equivariance in regularly gridded data.
> > > > > We argue then that the success of CNNs stems from the underlying principles of locality and equivariance (weight-sharing), and not the specific implementation for gridded data. Indeed, a lot of research has gone into overcoming the limitation in applicability of classical CNNs, which is a result of it not respecting the underlying continuous nature of spatial data and hence only being applicable to regularly gridded data. Consider for example PointConv [2], CKConv/CCNN [3], which attempt to generalize CNN architectures to point-cloud and irregular data, Spherical CNNs [4], which attempt to generalize CNNs to spherical domains, or Graph Convolutional Networks and Geometric Message Passing Networks [5, 6], which attempt to generalize the CNN architecture to (geometric) graph data. We feel these works should be seen as data type specific implementations of the same notions of locality and weight-sharing, to overcome the limitations placed on the applicability of the original CNN which, for no other good reason than ease of implementation, chooses to define convolution operators and the kernels themselves over a regular grid.
> > > > > ENFs retain exactly the desirable properties that explain the success of CNNs on regular spatial data; locality and equivariance (weight-sharing), but attempt to decouple this from the specific grid or domain on which the data is observed. The benefits to transferability that follows are immediate and shown in our experiments; we use the same downstream model for image, 3D point clouds and spherical data successfully.
> > > > >
> > > > > **To conclude**, we do not oppose including data-type specific architectural considerations in model design (making a specific choice of equivariance constraint as is done in each of our experiments is an example of a data-type specific constraint). Instead, we posit that decoupling architectural considerations (e.g. locality/weight-sharing) from the specific grid/geometry over which the data is observed is a desirable property, as it overcomes the need for adapting an architecture to a new data type only to account for such implementational problem parameters (as is done in e.g. PointConv, spherical CNNs, Geometric MPNNs).

---

> > > > > > ### Comment · Reviewer_9X4V · 2024-12-02
> > > > > >
> > > > > > Thank you for the calarifcations!
> > > > > >
> > > > > > **Resolution agnostic benefit**. The authors state that "...convey that the discretization-free nature of NeFs, and their resulting resolution agnosticity, is beneficial in real-world data settings.
> > > > > >
> > > > > > From my understanding, the suggested paradigm can be summarized as follows:
> > > > > > - Pretrain on a discrete signal dataset by reconstructing a continuous signal.
> > > > > > - Leverage the learned features in downstream tasks.
> > > > > >
> > > > > > In contrast, a "simpler" approach could be:
> > > > > > - Pretrain on a discrete signal dataset by directly reconstructing the discrete signal.
> > > > > > - Use the learned features in downstream tasks.
> > > > > >
> > > > > > While I acknowledge the theoretical potential benefits of the first approach (as well as some theoretical drawbacks previously discussed), I find the experimental evidence in the paper lacking a direct comparison or convincing argument regarding this simpler alternative. It leaves me uncertain about the intended practical takeaway.
> > > > > >
> > > > > >
> > > > > > **Transferability and task-specific architectures**. The authors state that "Instead, we posit that decoupling architectural considerations (e.g. locality/weight-sharing) from the specific grid/geometry over which the data is observed is a desirable property, as it overcomes the need for adapting an architecture to a new data type only to account for such implementational problem parameters".
> > > > > >
> > > > > > While I understand the proposed benefit, I find it challenging to fully grasp the issue with adapting an architecture to a new data type, especially if doing so proves beneficial. In my view, this seems more like a tradeoff than an unequivocal advantage.

---

> > > > ### Author Response · Authors · 2024-11-28
> > > >
> > > > Dear reviewer, we appreciate your effort and insightful comments in the discussion phase. If time permits, we kindly request you to give your opinion and insight on the responses we posted to your further concerns, such that we can further improve the strength of our work. Kind regards.

---

> ### Author Response · Authors · 2024-11-26
>
> **Geometry separation** The reviewer notes that our argument regarding the absence of the ability to separate pose and context information within the latent space of NeF literature is confusing, particularly in light of works [7, 8]. These works also introduce a steerability constraint, similar to ours, and utilize invariant and equivariant features. Although these works both include an experiment involving the use of an equivariant encoder-decoder architecture in implicit shape representation tasks, replacing the encoding in occupancy networks [9], neither of these works position themself as part of NeF literature (i.e. general continuous data representations). We acknowledge that the phrasing we used in our rebuttal was confusing and that indeed both of these works introduce (implicitly) the notion of a steerable implicit latent representation.
> We would like to stress that this phrasing appears only in our response to the reviewer and was not included in the manuscript itself. We change this phrasing in our response.
>
>
> [1] LeCun, Y., Boser, B., Denker, J., Henderson, D., Howard, R., Hubbard, W., & Jackel, L. (1989). Handwritten digit recognition with a back-propagation network. Advances in neural information processing systems, 2.
>
> [2] Wu, W., Qi, Z., & Fuxin, L. (2019). Pointconv: Deep convolutional networks on 3d point clouds. In Proceedings of the IEEE/CVF Conference on computer vision and pattern recognition (pp. 9621-9630).
>
> [3] Romero, D. W., Kuzina, A., Bekkers, E. J., Tomczak, J. M., & Hoogendoorn, M. (2021). Ckconv: Continuous kernel convolution for sequential data. arXiv preprint arXiv:2102.02611.
>
> [4] Cohen, T. S., Geiger, M., Köhler, J., & Welling, M. (2018). Spherical cnns. arXiv preprint arXiv:1801.10130.
>
> [5] Gilmer, J., Schoenholz, S. S., Riley, P. F., Vinyals, O., & Dahl, G. E. (2020). Message passing neural networks. Machine learning meets quantum physics, 199-214.
>
> [6] Satorras, V. G., Hoogeboom, E., & Welling, M. (2021, July). E (n) equivariant graph neural networks. In International conference on machine learning (pp. 9323-9332). PMLR.
>
> [9] Mescheder, L., Oechsle, M., Niemeyer, M., Nowozin, S., & Geiger, A. (2019). Occupancy networks: Learning 3d reconstruction in function space. In Proceedings of the IEEE/CVF conference on computer vision and pattern recognition (pp. 4460-4470).

---

### Official Review · Reviewer_yjcq · 2024-11-03

**Soundness:** 4
**Presentation:** 4
**Contribution:** 4
**Rating:** 8
**Confidence:** 3

**Summary:**

The paper introduces a novel class of Conditional Neural Fields (CNFs) called Equivariant Neural Fields (ENFs) which aim to address the limitations of CNFs in tasks requiring geometric reasoning. The authors propose a geometry-informed cross-attention mechanism that conditions on a latent point cloud of features, enabling equivariant decoding from the latents to the field of interest. This approach possess a steerability property where transformations in the field and mirrored in the latent space. Further, this approach ensures that the cross-attention attention operators respond similarly regardless of pose allowing for weight sharing over similar local patterns leading to more efficient learning. These claims are backed with experiments that demonstrate the advantages posed by the formulation and show a clear advantage over the baselines that have a geometry-free latent space.

**Strengths:**

- The paper introduces a novel and mathematically sound method to incorporating geometric structure to neural fields through the equivariant cross attention. The steerability property is well formulated with proven bi-invariant constraints.
- The experimental details demonstrate an advantage over methods that do not incorporate such geometry informed structure in the latent space. Additionally, the locality and weight-sharing properties discussed are clearly demonstrated.
- The paper is well-written providing clear background on the neural fields, and the motivation for the need for enforcing equivariance in neural fields. The diagrams are informative and highlight the key components of the methodology. Highlighing geometry attributes in Section 3 with a blue text color was particularly helpful in aiding understanding

**Weaknesses:**

- While the motivation to compare against other CNF based approaches is clear, the methodology seems to be restricted to a discussion and comparison to the results reported in functa (Dupont et al.) and other CNF-based methods but do not provide a thorough comparison against other equivariant methods or other state of the art methods. Perhaps a comparison of ENFs against more comparisons would strengthen the paper.

**Questions:**

- I'm particularly curious about the use of these equivariant neural fields as a general backbone for any neural field based task? Are there any situations where it's not helpful to enforce equivariance especially for vision / PDE-based applications?
- Have you considered using this methodology in a generative context? I think the localized latent point clouds are a particularly interesting property that could lead to more structured creation.
- Did you study the sample efficiency of ENFs against other CNF methodologies in tasks such as classification? One would assume that enforcing equivariance should lead to a better sample efficiency throughout all truncations of the training dataset
- I'm curious about the computational cost of your experiments. Does it have a similar run time to the other baselines that were discussed?

Additionally, I believe there are a couple of typos that I may have spotted:
- In the abstract: faitfhully -> faithfulll
- Also, on line 103, posses needs to be possess?

---

> ### Author Response · Authors · 2024-11-22
> **Initial response to reviewer yjcq**
>
> We thank the reviewer for their comprehensive evaluation of our manuscript. We appreciate the kind words about the clarity and novelty of the work and their excitement about using equivariant neural fields as general backbone for downstream tasks.
>
> **Baseline comparisons to other equivariant methods** The reviewer raised a valid weakness regarding not comparing to other equivariant baselines. We pose a general framework for acquiring continuous latent representations via CNF encodings for different data modalities and geometries. Therefore, we only compared methods which are applicable to different types of data modalities and geometries as well. Functa [1] is the original paper proposing this learning over arbitrary functasets and, as far as we know, no equivariant works exist in this line of research. There is a line of recent work that explores the use of deep weight-space methods (also referred to in our response to Rev. 4oVU) for amongst other tasks, learning over Neural Field representations, but we argue that due to the widely different scope and applicability of Conditional Neural Fields and weight-space methods (weight-space methods can be applied to general neural network architectures for tasks like model characteristic prediction [2], Functa / Conditional Neural Fields are generally applied to represent spatial signal data), this comparison is not sensible; highest-performing weight-space methods generally achieve around 45%-65% test set accuracy on CIFAR10 classification with very specific use of augmentations, and as such we argue that comparison to these approaches is ineffectual and might be confusing (see elaboration under ‘Comparison to weight-space methods’ in response to reviewer 4oVU). The current work is mainly interested in adding explicit geometry in the latent-spaces of CNFs encodings, and hence Functa is our main point of reference.
>
>
>
>
> The reviewer also raised some questions which we will answer below.
>
> **Application to non-equivariant settings** The reviewer raises the question whether there are scenarios where enforcing equivariance might not be beneficial. It could be argued that equivariance constraints limit the expressivity of the framework when applied to tasks that lack the same symmetries. However, we like to argue that adding the constraints, which enable weight-sharing, could still be beneficial in such a scenario. For instance, when analyzing the classification results of CIFAR-10 or ShapeNet16 there could be observed that even though these datasets do not exhibit exact symmetries within the data, translation equivariance -- via relative positions between latent points and sampled coordinates -- outperforms the same model using absolute positions. This suggests that weight-sharing over patches, derived from these relative relationships, leads to better performance. However, you could also observe from the same experiment that restricting it further (to SO(2) equivariance) did harm the performance a bit. So we posit that restricting the model too much could still be harmful when the symmetry is not contained in the data.

---

> ### Author Response · Authors · 2024-11-22
> **Second part of initial response to reviewer yjcq**
>
> **Generative modelling on ENFs** The reviewer raises the question whether we considered applying our methodology in the generative context. This is a very interesting possible application of ENFs, and as such, we follow [1], and train a diffusion model. With this rebuttal we also add an extra experiment on generative modeling over ENF latent spaces in Tab. 7, Fig. 9. We train a diffusion model, parameterized as a Diffusion Transformer (DiT-B)--a natural choice due to the set-structure of our ENF latent space–on the latents that we obtain from pretraining the ENF on an image reconstruction objective. We first obtain a set of latents for each image, and subsequently use these images in a denoising objective, and then train a DiT-B on a denoising diffusion objective on this latent space, where the forward diffusion kernel is given by:
>
> $$z_t = \big\[(p_{i}, \sqrt{\bar{\alpha}_t} \mathbf{c}_i^0 + \sqrt{1-\bar{\alpha}_t} \epsilon^{\mathbf{c}})\big\]^N_i$$
>
> with $\epsilon^\mathbf{c} \in N(0, 1)$, and subsequently sample using DDIM. Details are added to Appx. C4. Notably, results in Tab. 7 and Fig. 9, show that although ENF and Functa perform comparably in terms of FID on CelebA (33.8 and 40.4 FID respectively), only ENF is able to generalize to non globally-aligned image data, obtaining 23.5 FID on CIFAR10, compared to 78.2 FID obtained by Functa. Visually (see Fig. 9), these results are corroborated, and ENF produces crisper samples compared to Functa. For comparison, we also add baseline results for other field-based generative models (both latent and explicit field parameterizations), but note that all of these models were trained on a generative objective, whereas in the case of Functa and ENF the generative process is trained on top of the latent space of a self-supervised pre trained Neural Field (i.e. no access to the image data is needed during the training of the generative process). These observed results align with the reviewer and our intuition that including explicit geometry is structuring the generations and improving the generative capabilities, and future work could further explore generative adaptations of ENF for better performance or broader application.
>
> **Sample efficiency of ENF** The reviewer asked whether we evaluated the sample efficiency of our method compared to the Functa baseline. Although we did not do a full evaluation on this part, we think that our flood-map segmentation results on the Ombria dataset (table 4) indicate the improved sample efficiency of ENF compared to Functa. The Ombria dataset is a small dataset containing only 800 training samples. Functa achieves a decent reconstruction PSNR and IoU of 31.5 and 93.7 respectively on the training set, however, achieves only a PSNR and IoU of 16.8 and 42.8 respectively on the test set. On the contrary, ENFs were able to generalize given this small training set achieving a PSNR and IoU of 31.6 and 74.0 respectively on the test set. We believe this is a sign for higher sample efficiency of ENF compared to Functa, which is observed across equivariance literature.
>
> **Details on computational efficiency** Lastly, the reviewer asks about the computational costs of the proposed method. Since another reviewer, AKXe, also requested this information, we want to refer to the table provided in our response to AKXe. The table shows that even without the kNN efficiency trick, ENF is more efficient in terms of FLOPs and seconds per epoch. Memory usage is the main bottleneck; however, applying the efficiency trick resolves this issue resulting in a much lower memory GPU usage than Functa. This information is added to the revised appendix.
>
> We would like to thank the reviewer again for their valuable suggestions and questions, and would like to invite the reviewer to discuss if any concerns remain.
>
> [1] Dupont, E., Kim, H., Eslami, S. M., Rezende, D., & Rosenbaum, D. (2022). From data to functa: Your data point is a function and you can treat it like one. arXiv preprint arXiv:2201.12204.
> [2] Schürholt, K., Kostadinov, D., & Borth, D. (2021). Self-supervised representation learning on neural network weights for model characteristic prediction. Advances in Neural Information Processing Systems, 34, 16481-16493.

---

> > ### Comment · Reviewer_yjcq · 2024-11-28
> > **Questions Addressed**
> >
> > I appreciate the authors' thoughtful efforts in addressing my questions and concerns. I am glad to see that most of my questions have been addressed with additional experiments for the generative modelling as well. I will maintain my original score and continue to recommend your paper for acceptance. Thank you for your detailed responses and clarifications!

---

### Official Review · Reviewer_AKXe · 2024-11-03

**Soundness:** 3
**Presentation:** 3
**Contribution:** 3
**Rating:** 8
**Confidence:** 3

**Summary:**

The paper introduces the Equivariant Neural Fields, a variant of conditional neural field that uses a geometry-informed cross-attention to condition the NeF using geometrical point cloud representation. The method was validated using a variety of applications, including classification, segmentation, forecasting, and reconstruction.

**Strengths:**

***Clear and Professional Presentation***: The paper is well-written, structured effectively, and easy to follow. Its clear motivation, logical organization, and high-quality visualizations contribute to a polished and professional presentation, making the methodology accessible and engaging.

***Introduction of Equivariant Neural Fields Model***: The authors propose a novel model, Equivariant Neural Fields, which combines conditional neural fields with point cloud conditioning and equivariant decoding from latent space to field. This approach creatively integrates Neural Fields with equivariant models designed for point clouds, expanding on existing techniques. Additionally, the paper introduces specialized attention layers and engineering optimizations that enhance the model's efficiency, showcasing an innovative blend of established methods.

***Comprehensive Experimental Validation***: The method is rigorously tested across a wide range of use cases and downstream tasks spanning various domains. This extensive evaluation demonstrates the versatility and potential real-world applicability of the proposed approach, supporting its robustness and utility across diverse applications.

**Weaknesses:**

***High Time Complexity***: The proposed approach appears to be computationally intensive. It would be beneficial for the authors to compare the training time and memory usage of their method against a reference model, such as the Functa method, to provide a clearer assessment of its efficiency.

***Lack of Ablation Studies***: The paper would benefit from ablation studies to clarify the contributions of key components, such as Gaussian spatial windowing and the k-nearest neighbors (kNN) efficiency trick. These studies would help demonstrate how each element impacts the model’s training efficiency and overall performance.

***Suboptimal Segmentation Performance****: The segmentation results are weaker than those of traditional point cloud segmentation baselines. A deeper investigation and discussion of these performance differences would help in understanding and potentially addressing the gaps in segmentation accuracy.

**Questions:**

Please refer to the weakness section

---

> ### Author Response · Authors · 2024-11-22
> **Initial response on reviewer AKXe**
>
> We thank the reviewer for their thorough assessment of our work, and are glad to see that the reviewer appreciates the clarity of our presentation, as well as the novelty of the proposed method and rigor of our experimental validation.
>
> The reviewer raises a number of valid concerns, which we address below.
>
> **High time complexity** The reviewer rightfully raises concerns regarding computational complexity of our method. Indeed, as noted in the manuscript, the original formulation of the ENF architecture is computationally expensive due to the “global” computation of attention coefficients. However, using the kNN approximation we’re able to drastically lower computational complexity; computational complexity for the original method scales quadratically with the number of latents N and input coordinates M; O(N*M), whereas when applying the kNN approximation, we can keep K fixed to a relatively small number (e.g. k=4, leading to O(4M) complexity). We recognize the need for quantitative evaluation of the computational efficiency of our method, and as such add an ablation to Appx. D. Below we show the reduction in runtime that the kNN approximation yields, as well as a comparison with Functa. These results show that ENF with kNN approximation uses similar number of FLOPs and memory compared to Functa, but is significantly (>10x) faster in runtime, which we attribute to the shallow nature of our ENF model as compared to the relatively deep SIREN used in Functa. Additionally, we investigate the impact that the kNN approximation has on performance below (also added to Appx. D.), and show that impact to both reconstruction and downstream performance are negligible.
>
> | model| Flops ($\times$10^9) | Memory (Gb/sample) | Time per epoch (s) |
> |-|-|-|-|
> | Functa | 28.3 | 0.6125  | 2864|
> | ENF (no kNN)  | 104.5| 1.825| 1801|
> | ENF (kNN, k=4) | 22.7| 0.400| 207|
>
> **Lack of ablation studies**
> The reviewer highlights that the manuscript would benefit from adding ablation studies on the Gaussian spatial windowing (GSW) and the k-nearest neighbors (kNN) approximation. We do agree with this and provide the ablation study in the table below. We update this rebuttal, now that we've finalised running these experiment on CIFAR-10 classification. We keep the hyperparameters identical to the ones listed for Tab. 1 / Sec. 4.2, but notably do not train the downstream model on augmented CIFAR-10, and train the ENF backbone for only 10 epochs.
>
> |model|Test PSNR|Test Acc|
> |-|-|-|
> |ENF w/o GSW/KNN| 38.8|71.0|
> |ENF + kNN|39.2|71.8|
> |ENF + GSW|39.5|74.2|
> |ENF + GSW + kNN|39.9|75.0|
>
> As can be seen, although reconstruction performance is relatively unaffected by ablating over the kNN and GSW (i.e. when removing them from the model), the downstream performance is significantly lower when not using GSW. This is in line with our expectations, as (like explained in Sec. 3.1) there is nothing enforcing locality of the latents. Using only the kNN slightly improves performance, possibly since it adds some measure of locality back into the latent space, but it seems the smoothness in locality enforced by the GSW is vital for downstream performance. The Gaussian spatial windows significantly improve the performance on CIFAR-10 classification and reconstruction, since it allows for weight-sharing between latents.
>
> **Suboptimal Segmentation Performance**
> We acknowledge the reviewer's observation that our segmentation results are weaker compared to traditional point cloud segmentation baselines. Since other reviewers (9x4v, 4oVU) also identified this as a weakness, we choose to de-emphasize this experiment to the appendix and in turn, follow [1] in adding an experiment showcasing generative modelling capabilities of the ENF framework in the main text (see also our response to Rev. yjcq). The segmentation experiment was initially included because it showcases versatility of Neural Fields in varying datatypes and modalities. We find that these findings– due to the global alignment of ShapeNet data– actually demonstrate that our method performs comparable to baselines in settings without global symmetries. As discussed in Appendix D.2, we found that without any conditioning on the acquired latents and using only the class embedding, ENF achieved class and instance mIoU scores of 64.3 and 69.2, respectively. This indicates that many points in this dataset can be correctly segmented purely based on their absolute positions. However, we believe the generative modeling experiment more effectively highlights the advantages of our method. Nonetheless, we will retain the segmentation experiment in the appendix for reference, as they indeed showcase that equivariance as inductive bias has limited benefit in settings without global symmetries.
>
> We are happy to continue the discussion when the concerns are not completely addressed yet.
>
> EDIT: We updated the results for the ablation over KNN/GSW in the above table.

---

> > ### Author Response · Authors · 2024-11-22
> > **Reference list of our response to reviewer AKXe**
> >
> > [1] Dupont, E., Kim, H., Eslami, S. M., Rezende, D., & Rosenbaum, D. (2022). From data to functa: Your data point is a function and you can treat it like one. arXiv preprint arXiv:2201.12204.

---

> > > ### Comment · Reviewer_AKXe · 2024-11-26
> > >
> > > Dear Authors, thank you for addressing my concerns, especially those related to efficiency and initial results for ablation studies. I decided to rise the score.

---

> ### Author Response · Authors · 2024-11-26
>
> Dear reviewer, thanks again for helping us to improve the manuscript, your suggestions and feedback were valuable!!

---

### Official Review · Reviewer_4oVU · 2024-11-04

**Soundness:** 2
**Presentation:** 2
**Contribution:** 2
**Rating:** 6
**Confidence:** 4

**Summary:**

This paper proposes equivariant conditional neural fields based on steerable networks. Architecture-wise, this paper proposes equivariant  cross-attention layers with Gaussian windowing as the basis of their Equivariant Neural Fields (ENF). The ENF is trained with a two-stage process: in the first stage, the ENF backbone takes in an input signal and outputs a latent point cloud of (pose, context) pairs. Downstream tasks can be accomplished by training a decoder which takes the latent point cloud as input. Experiments are performed on 2D image reconstruction and classification, 3D reconstruction, classification, and part-segmentation, flood map segmentation, and climate forecasting.

**Strengths:**

This paper creates a novel equivariant neural field based on the notion of steerability from equivariant networks, which has the advantages of weight-sharing, locality, and geometric interpretability.

**Weaknesses:**

1. The original Functa paper uses a SIREN neural field architecture but this paper uses an attention-based neural network architecture. This seems like a potentially unfair comparison.
2. Another weakness of this paper is that there is no way to decide ahead of time whether to train the latent point cloud using MAML or autodecoding.
3. The only baseline is Functa for most experiments. Is it possible to use NF2vec in Table 2 and Inr2Array [1] for any of the experiments involving downstream tasks? For tasks involving generalization,
4. ENF performs only comparably to to the baselines on part-segmentation (Table 3), and some experiments (Table 2) don't show the effectiveness of using equivariance.

**Questions:**

1. Does it make sense to compare against Inr2Array [1]?
2. Should NF2vec also be a baseline for the shape classification task (Table 2)?
3. Can Functa be trained with a cross-attention-based architecture, similar to that proposed for ENF?
4. For Functa baselines on downstream tasks such as classification, what was the architecture of the decoders used?

[1]: Zhou, Allan, et al. "Neural functional transformers." Advances in neural information processing systems 36 (2024).

---

> ### Author Response · Authors · 2024-11-22
> **Initial response to reviewer 4oVU.**
>
> We thank the reviewer for taking the time to evaluate our manuscript thoroughly and contributing to its improvement. We address each of the reviewers' concerns separately below. We hope to continue the discussion if any concerns remain.
>
> **Difference in architecture compared to Functa baseline** The reviewer highlights the contrast between the proposed ENF architecture and the Functa [1] architecture used as baseline - built on top of SIREN [2]. We assert that comparing Functa with our attention-based architecture is a reasonable and relevant comparison, as it remains the most prominent work on CNF-based signal representations, making it an essential point of reference for our proposed model. Functa introduces a specific framework for parameterizing signals through layer-wise MLP shift modulations, parameterized by a latent vector. This latent vector may then be used by simple MLP-based downstream models. Because this single vector constitutes the conditioning variable in Functa, it is not possible to use a cross-attention operation in combination with the original Functa architecture, and in fact one of our primary contributions lies in proposing a method for utilizing point clouds as conditioning variables in CNFs. To enable additional comparison, we provide specific parameter counts, inference time and memory complexity of our model compared to Functa in our response to Rev AKXe, showing drastically improved parameter efficiency of our model which we attribute to the inclusion of locality and weight-tying as inductive biases.
>
> We do agree (also noted by Rev. AKXe ) that an additional ablation over the specific geometric conditioning that we propose may contribute to better interpretability of the experimental results, and as such we perform an additional experiment where we use a geometry-free latent conditioning set, i.e. the latents have no position and as a result no locality (this approach can be seen as an extension of 3DShape2VecSet from shape to arbitrary signal data). We achieve this by making the “bi-invariant” $\mathbf{a}$ (and as a result $\mathbf{q}$) only a function of $x$. We find that this implementation of the framework –keeping all hyperparameters identical to the setup we used for CIFAR classification – leads to highly unstable training that saturates around 22 reconstruction PSNR on the test set, likely attributable to the fact that now any update to one of the latent codes affects the output of the NEF globally, leading to a much more complex optimization landscape. This highlights another advantage of either having a single global latent, or using locality as inductive bias; optimization of single or locally responsible latents seems to lead to a simpler optimization landscape compared to optimizing a set of global latents. We apply a simple transformer with 4 layers, 256 hidden dim and 4 heads as a downstream classifier (without positional encoding, since the latents don’t have positional attributes in this setting). We train for 500 epochs with Adam using a learning rate of 1e-4, after which the train loss has converged, obtaining 0.98 train and 0.43 test set accuracy. We observed overfitting early into training. Utilizing early stopping, best performance was achieved after just 5 epochs, yielding a test set accuracy of 0.47. These observations are in line with the outcome of our other experiments; geometry-grounded latents are more informative for downstream tasks. The table below is added to Appx. D.
> | CIFAR10|Recon (test PSNR)|Class (test acc)|
> |-|-|-|
> |ENF w/ ${\mathbb{R}^2}$ latents|42.2|82.1
> |ENF w/ pose-free latents|22.3|47.9
> |Functa|38.1|68.3
>
> **What downstream Functa architecture is used** We recognize the need for reproducibility, and added more info on the Functa baseline used in Appx. C.2 to include details on the downstream architectures we used. We’ve edited the main body of the text to refer to this section more clearly. We tried keeping close to the original setup used in [1], and so for the downstream architectures we use a 3-layer 1024 hidden dim residual MLP, slightly larger in parameter count compared to the 3-layer 256 hidden dim PONITA MPNN that we use in all our ENF experiments (2.1M vs. 1.7M). Like in our ENF experiments, the same architecture is used in classification, segmentation and forecasting, only the output head is changed.

---

> > ### Author Response · Authors · 2024-11-22
> > **Continuation of first response on 4oVU**
> >
> > **Using MAML or autodecoding** The reviewer highlights that it is not clear from the manuscript how to decide beforehand whether to use MAML or auto decoding to obtain the latents. In Appx. A.1.1, A.1.2, we expand on why we think meta-learning is always preferable compared to auto decoding, highlighting two main reasons; 1) it reduces inference time significantly, only requiring up to 3 SGD steps to fit a novel signal instead of ~200-500 [2], and 2) it implicitly regularizes the ENFs latent space by constraining modulation sets to lie within a small distance of the shared initialization [1, 3]. However, as observed in [1], using meta-learning to fit functasets has some notable limitations when fitting complex signals and [1] remark on the limited expressivity of Meta-Learning due to the small number of gradient descent steps used to optimize a latent. We observed similar performance limitations on shape data, possibly attributable to the sheer size of the point clouds / voxel grids operated on. We follow the advice of [1] and instead use auto decoding in this setting. In practice, choosing between MAML and auto decoding can be based on a simple hyperparameter search; if MAML does not perform well, use auto decoding.
> >
> > **Comparison to weight-space methods** The reviewer highlights that in most experiments we compare only to Functa, and asks whether comparison to methods like INR2Array[4] or INR2Vec/NF2Vec [5] are meaningful. We compare our method with Functa as it is the paper that originally introduced the concept of learning over functasets as a method for deep learning on continuous data, and in approach is the most similar method in literature to ours; i.e. encoding a signal through a latent that conditions a Neural Field and using this latent as a signal representation downstream. Notably, the methods referred to by the reviewer operate on weight-space; they are instances of frameworks that operate on non-conditional Neural Fields, where every signal is represented by an individual Neural Field (or even on non-field applications of neural networks, e.g. predicting generalization of a CNN [6]). Although it has broader applicability, in the context of Neural Fields specifically this approach has a number of notable drawbacks also noted in previous works [1, 2]; 1) it requires optimizing and storing a full neural network for every new signal (e.g. [5] show that reconstruction accuracy for smaller SIRENs –3 layers 32 hidden dim, ~2.2k parameters– is limited to around 25 PSNR even after 5k optimization steps, [6] represent a shape with a 4 layer 512 hidden dim SIREN or ~800k parameters, we instead fit CIFAR with 25 latents of size 32 or 800 parameters in 3 SGD steps). This approach quickly becomes infeasible for larger datasets and more complex data (e.g. fitting 50 augmentations to each CIFAR image as is done in [1] with the SIREN used in [6] would result in a parameter dataset of ~1.3TB). 2) performance in downstream tasks is severely limited due to complexities of operating on weight space arising from ambiguities and symmetries in this space. [4] reports test classification performance on CIFAR10 for INR2Vec [8] DWS [7], NFN [9] and INR2Array [4] (all weight-space methods), obtaining **16.7, 42.9, 46.6 and 63.4** test accuracy respectively – a very large gap to **82.4** we report with ENF. To us, the difference in objective and applicability between Functa and weight-space methods (learning over continuous signal parameterizations vs. learning over generic weight-spaces) makes comparison ineffectual and confusing.
> >
> > To conclude, although we would be happy to include baseline comparisons in the classification or other experiments, if the reviewer(s) deem this relevant, in our opinion these are not useful comparisons to make due to the drastic differences in complexity and applicability of weight-space methods and learning over functasets.
> >
> > We chose to compare with NF2Vec [10] in the ShapeNet segmentation experiments because this method is tailored specifically to representing continuous 3D shape data, and so it aligns better with our work, making comparison more useful. The reviewer suggests to then also compare with NF2Vec also for the ShapeNet classification task. We feel this is a good suggestion, and include classification results for NF2Vec [10] on voxelized ShapeNet (obtaining 93.3 compared to 96.6 w/ ENF).

---

> ### Author Response · Authors · 2024-11-22
> **Final part of the first response to reviewer 4oVU**
>
> **Weakness: comparable performance to baselines on ShapeNet segmentation**
> The reviewer points out that our segmentation results are weaker compared to traditional point cloud segmentation baselines. Noting that other reviewers (9x4v, AKXe) have also highlighted this issue, we intend to move this experiment to the appendix and replace it with a generative modeling experiment in the main text. We initially included the segmentation experiment because it lacks symmetries, thereby demonstrating that our method performs comparably even in non-symmetric settings. As detailed in Appendix D.2, we discovered that without conditioning on the acquired latents and using only the class embedding, ENF achieved class and instance mIoU scores of 64.3 and 69.2, respectively. This indicates that many points in this dataset can be correctly segmented based solely on their absolute positions. However, we believe that the generative modeling experiment more effectively highlights the strengths of our method. We will retain the segmentation experiment in the appendix for reference.
>
> We would like to thank the reviewer again for their valuable suggestions, and invite the reviewer to discuss if any concerns remain.
>
> [1] Dupont, E., Kim, H., Eslami, S. M., Rezende, D., & Rosenbaum, D. (2022). From data to functa: Your data point is a function and you can treat it like one. arXiv preprint arXiv:2201.12204.
>
> [2] Yin, Y., Kirchmeyer, M., Franceschi, J. Y., Rakotomamonjy, A., & Gallinari, P. (2022). Continuous pde dynamics forecasting with implicit neural representations. arXiv preprint arXiv:2209.14855.
>
> [3] Knigge, D. M., Wessels, D. R., Valperga, R., Papa, S., Sonke, J. J., Gavves, E., & Bekkers, E. J. (2024). Space-Time Continuous PDE Forecasting using Equivariant Neural Fields. arXiv preprint arXiv:2406.06660.
>
> [4] Zhou, A., Yang, K., Burns, K., Cardace, A., Jiang, Y., Sokota, S., ... & Finn, C. (2024). Permutation equivariant neural functionals. Advances in neural information processing systems, 36.
>
> [5] Papa, S., Valperga, R., Knigge, D., Kofinas, M., Lippe, P., Sonke, J. J., & Gavves, E. (2024). How to Train Neural Field Representations: A Comprehensive Study and Benchmark. In Proceedings of the IEEE/CVF Conference on Computer Vision and Pattern Recognition (pp. 22616-22625).
>
> [6] Zhou, A., Yang, K., Jiang, Y., Burns, K., Xu, W., Sokota, S., ... & Finn, C. (2024). Neural functional transformers. Advances in neural information processing systems, 36.
>
> [7] Navon, A., Shamsian, A., Achituve, I., Fetaya, E., Chechik, G., & Maron, H. (2023, July). Equivariant architectures for learning in deep weight spaces. In International Conference on Machine Learning (pp. 25790-25816). PMLR.
>
> [8] De Luigi, L., Cardace, A., Spezialetti, R., Ramirez, P. Z., Salti, S., & Di Stefano, L. (2023). Deep learning on implicit neural representations of shapes. arXiv preprint arXiv:2302.05438.
>
> [9] Zhou, A., Yang, K., Burns, K., Cardace, A., Jiang, Y., Sokota, S., ... & Finn, C. (2024). Permutation equivariant neural functionals. Advances in neural information processing systems, 36.
>
> [10] Ramirez, P. Z., De Luigi, L., Sirocchi, D., Cardace, A., Spezialetti, R., Ballerini, F., ... & Di Stefano, L. (2024). Deep Learning on Object-centric 3D Neural Fields. IEEE Transactions on Pattern Analysis and Machine Intelligence.

---

> > ### Comment · Reviewer_4oVU · 2024-11-27
> >
> > Thank you for addressing my concerns, especially those regarding comparisons and baselines. I will raise my score.

---

### Author Response · Authors · 2024-11-25
**Invitation to participate in further discussion**

Dear Reviewers,

Thank you once again for your thoughtful reviews and constructive feedback on our manuscript. We appreciate the time and effort you’ve put into evaluating our work.

As we approach the end of the discussion period, we would like to kindly invite you to participate in the ongoing discussion. We feel we have addressed the points raised in your reviews, including additional experiments (Rev. yjcq), ablations (Rev. 4oVU), information and comparison on computational efficiency (Rev. yjcq, AKXe), and clarifications and revisions to our manuscript to clarify motivation and claims made in our work (Rev. 9X4V). Your insights have significantly improved the quality and rigor of our work, and we want to ensure that any remaining concerns are addressed thoroughly.

We would greatly value your engagement in this discussion to confirm that we’ve adequately addressed your comments, and invite you to amend your recommendations as you see fit. Additionally, we would like to explore any additional suggestions you might have; if there are specific areas where you feel further elaboration or revision is needed, please do not hesitate to share your thoughts.

Thank you for your continued involvement. We look forward to your response.

---

### Meta-Review · Area_Chair_jt8k · 2024-12-22

**Metareview:**

This paper proposes a variant of conditional neural fields by utilizing a latent point cloud rather than a global latent for a neural field. The authors claimed locality and steerability for the resulting Equivariant Neural Fields and demonstrated the effectiveness of latent for many tasks including perception and generative modeling. This paper is well-written as acknowledged by multiple reviewers. The implementation of equivariance through cross-attention between queries and latents is simple and elegant. Experiments clearly demonstrated the usefulness of geometry equivariance for representation learning.
The downside of the proposed method is the extra computational complexity of quadratic cross-attention between queries and multiple latents in a neural field. The authors have proposed a based approximation to full attention operation.
All four reviewers gave positive ratings for this paper. The AC recommends acceptance of this paper due to the novelty in conditioning neural fields on latent point cloud.

**Additional Comments On Reviewer Discussion:**

Multiple reviewers (9x4v, 4oVU, AKXe) were concerned about the inferior performance for segmentation tasks.
The authors have acknowledged the limitation of the proposed method and further added a new task of generative modeling which better shows the advantage of the proposed conditioning mechanism.

Reviewers were also concerned about high computational complexity with cross attention between query and latent point cloud.
The authors have shown effective acceleration with KNN-based attention.

---

### Decision · Program_Chairs · 2025-01-22

Accept (Poster)